# A model for the Artic mixed layer circulation under a summertime lead: Implications on the near-surface temperature maximum formation

Alberto Alvarez

NATO-STO Center for Maritime Research and Experimentation-CMRE, V. San Bartolomeo 400, La Spezia, 19126, Italy

*Correspondence to*: Alberto Alvarez (alberto.alvarez@cmre.nato.int)

**Abstract.** Leads in sea ice cover have been studied extensively because of the climatic relevance of the intense ocean–atmosphere heat exchange that occurs during winter. Leads are also preferential locations of heat exchange and melting in early summer, but their oceanography and climate relevance, if any, remain largely unexplored during summertime. In

particular, the development of a near-surface temperature maximum (NSTM) layer typically 10–30 m deep under different Arctic basins has been related observationally to the penetration of solar radiation through the leads. These observations reveal that the concatenation of calm and wind events in the leads could facilitate the development of the NSTM layer. Using numerical modelling and an idealized framework, this study investigates the formation of the NSTM layer under a summer lead exposed to a combination of calm and moderate wind periods. During the calm period, solar heat accumulates in the upper

layers under the lead. Near-surface convection cells are generated daily, extending from the lead sides to its centre. Convection cells affect the heat storage in the mixed layer under the lead and the adjacent ice cap. A subsequent wind event (and corresponding ice drift) mixes and spreads fresh and cold meltwater into the warm layers near the surface. Surface mixing results in temperatures in the near-surface layers that are lower than in the deeper layers, where the impact of the surface stresses is weaker. Additionally, the warm waters initially located under the lead surface stretch and spread horizontally. Thus,

an NSTM layer is formed. The study analyses the sensitivity of depth and temperature of the NSTM layer to buoyancy forcing, wind intensity, ice drift, stratification and lead geometry. Numerical results suggest that the NSTM layer appears with moderate wind and ice drift and disappears when the wind intensity is higher than 9 ms$^{-1}$. Depending on the background stratification, the calm period reinforces or becomes critical in NSTM layer formation. According to the results, ice drift is key to the development of the NSTM layer.


## 1 Introduction

The Arctic Ocean is currently responding to changes in atmospheric and oceanic processes associated with climate change by

a rapid retreat of sea ice (Liu et al., 2012; Stroeve et al., 2007). Ice retreat not only affects the local Arctic environment but also has feedback on the global climate. Ice melting might influence climate components such as the increase in abnormal weather events in the Northern Hemisphere and modifications in the stability of the thermohaline circulation, among others (Levermann et al., 2007; Vihma, 2014).

Sea ice loss involves heat transfers in the ice–atmosphere–ocean system mediated by physical mechanisms such as atmospheric

and oceanic heat transport (Zhang et al., 2008; Spielhagen et al., 2011), variations in water vapour and cloudiness (Schweiger et al., 2008), or modifications in sea ice cover (Screen and Simmonds, 2010). In particular, cracks in the sea ice cover (also known as leads) are ice-free areas where strong heat and chemical exchanges occur between the ocean and the atmosphere (Maykut, 1978; Alam and Curry, 1995; Douglas et al., 2005; Kort et al., 2012; Steiner et al., 2013). Leads are typically

generated by stress-deformation events in continuous sea ice cover. These events develop elongated, recurring structures of

open water and thin ice. Their geometry ranges from 10 m to 1 km wide and up to 100 km long (Wilchinsky et al. 2015). Due to their climate relevance, a number of studies have considered the identification and characterization (width, orientation, area of coverage and spatial distribution) of leads in ice cover (Barry et al., 1989; Miles and Barry, 1998; Brohan and Kaleschke, 2014; Wernecke and Kaleschke, 2015; Hoffman et al. 2019).

In winter, leads rapidly refreeze with frazil ice production due to large heat fluxes from a relatively warm ocean exposed to

cold air. The refreezing of a lead produces a buoyancy flux by brine rejection, which generates a particular circulation pattern at the lead location: the dense water flows away from the lead when it reaches the bottom of the mixed layer, while freshwater flows in from the lead sides near the surface (Kozo, 1983; Morison et al., 1992; Morison and McPhee, 1998). Salt rejected by brine constitutes a major salt input to the Arctic mixed layer (Morison and Smith, 1981; Lemke and Manley, 1984; Morison et al., 1992).

During the early melting season, refrozen leads become preferential melting sites (Perovich et al., 2001). This is because the new ice in the leads is thinner and has a lower albedo than the adjacent ice (Grenfell and Maykut, 1977; Tschudi et al., 2002). Additionally, lead locations are topographically lower, accumulating meltwater and further reducing albedo. Perovich et al. (2001) found that ice in the leads melts almost completely by the end of July. At this time, the ice landscape is made up of large recurring sea ice slabs separated by long, narrow open leads. As the thermal deterioration of the sea ice progresses, this

landscape transforms into a complex mosaic of sea ice sheets intertwined with open waters (Perovich et al., 2001).

The development of a near-surface temperature maximum (NSTM) layer typically 10–30 m deep under different Arctic basins has been attributed to the penetration of summer solar radiation through leads (Maykut and McPhee, 1995; Jackson et al., 2010; Kadko, 2000). The temperature in this layer is usually more than one or two tenths of a degree above the underlying temperature or the freezing point, respectively (Jackson et al., 2010; Steele et al., 2011). An NSTM layer would result from

solar heating beneath a protective halocline generated by fresh water accumulated by melting sea ice. Strong near-surface stratification preserves the NSTM layer for long periods, likely affecting the oceanographic structure and acoustic properties of the upper ocean and overlying sea ice cover. For example, the NSTM layer could significantly enhance sea ice melt from turbulent heat fluxes to the basal sea ice caused by wind or ice motion (Ramudu et al., 2018). Sound ducts with a significant capacity for long-range acoustic propagation may also result from the formation of the NSTM layer (Freitag et al., 2012).

Some observational evidence correlates the development of the NSTM layer with a combination of buoyancy and wind events in the leads (Gallaher et al., 2017). Under quiescent conditions, a warm layer of fresh water on the surface of the summer leads is the result of the combined effect of meltwater runoff and solar heating (Richter-Menge et al., 2001; Gallaher et al., 2017). This warm freshwater layer deepens and spreads laterally under adjacent ice if calm conditions persist (Richter-Menge et al., 2001). Wind events after a calm period induce further mixing of the warm freshwater layer with the underlying ocean when

wind and/or sea ice speeds reach a threshold (Richter-Menge et al., 2001). Wind and/or sea ice drift erode and deepen the warm freshwater layer in the summer lead. However, the summer halocline protects the warm layers from the action of wind and sea ice (Gallaher et al., 2017). Therefore, a sequence of calm periods and moderate winds could develop an NSTM layer under the summer leads.

Numerical studies on leads in summer are scarce in the literature. Skyllingstad et al. (2005) provided insight into lateral melting

processes in a summer lead using a large-eddy simulation (LES) model. Their numerical study suggested that lateral melt rates increase with lead size but decrease in high winds due to turbulent mixing with cold waters residing beneath the sea surface. Ramudu et al. (2018) also used an LES model to numerically investigate the relevance of heat turbulent fluxes from a mixed layer on sea ice melt in a summer lead. Their results showed the heat entrainment from a warm layer near the surface that generally forms in summer at different Arctic locations. The author is not aware of other numerical studies in summer leads.

In particular, the development of an NSTM layer in a summer lead under a sequence of calm and windy conditions still remains numerically unexplored.

This study investigates the dynamics in a summer lead exposed to a sequence of calm and moderate wind conditions such as those reported in Richter-Menge et al. (2001) at the SHEBA ice station from July 18th to July 31st, 1998. An axisymmetric geometry and a particular thermodynamic forcing are common features of summer leads under calm conditions. For this

reason, the study initially focuses on the circulation under a summer lead resulting from the combined effects of lead geometry, solar radiation and sea ice melt. Under these conditions, lateral buoyancy gradients between the edges and centre of the lead due to solar heating and ice melt can trigger circulation in the lead. This circulation would result in a warm layer of fresh water on the surface of the lead. The study is then completed with an analysis of the mixing and deepening, if any, of the warm freshwater layer due to wind events (and associated sea ice drift) after the calm period. The ultimate goal of the study is to

assess whether these environmental conditions from field observations could result in the formation of an NSTM layer as inferred from the observations.

## 2 Methodology

### 2.1 The physical model

The conceptual framework considers a vertical cross-section of an ice landscape described by recurring and large sea ice slabs separated by long, narrow, rectilinear leads. The landscape was idealized here by a large (infinite) fringe pattern with sea ice slabs and open water leads 250 m and 50 m wide, respectively (Figure 1). Field observations reveal that the maximum occurrence of lead geometry corresponds to leads and refrozen leads less than 100 m wide (Barry et al., 1989, Tschudi et al., 2002) and distances between leads less than 500 m (Haggerty et al., 2003). Therefore, the selected geometry is representative

of the main lead fraction.

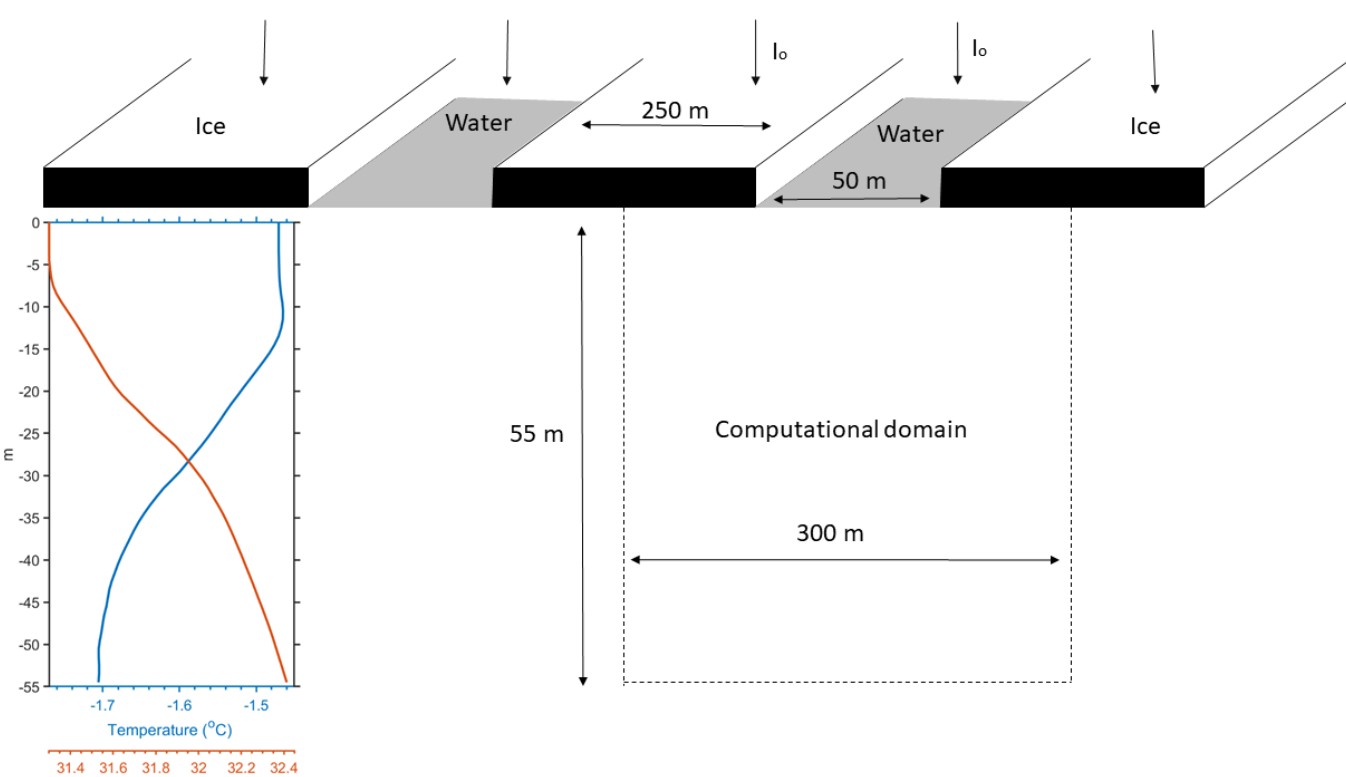

**Figure 1: Idealized framework of the current study, showing the incoming solar radiation, I$_o$, and initial profiles of temperature and salinity, respectively.**

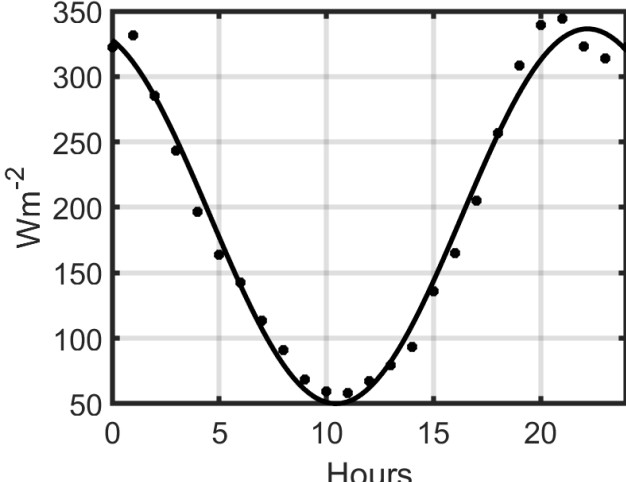

**Figure 2: Daily incoming shortwave radiation obtained by averaging records from July 18th to 26th, 1998 at SHEBA Ice Station (Persson et al., 2002). Averaged data is represented by the black dots. Solid black line fits the raw data to a periodic function with a least-squares method.**

The sea ice slabs are assumed to be stationary in the calm period and drift during the subsequent wind event. Reference temperature ($T_{ref}$) and salinity ($S_{ref}$) profiles were obtained by averaging daily conductivity, temperature and depth measurements (CTDs) collected at the SHEBA ice station from July 18th to July 26th, 1998 (Stanton and Shaw, 2006) (Figure 1). The winds were light during this period, and thus the stratification is considered representative of the calm stage. Reference fields only depend on depth. The current study focuses on the upper 55 m of the water column. Below this depth, an isothermal layer approximately 20 m thick was found, which facilitated the implementation of appropriate boundary conditions (see below).

Solar radiation is the only energy source considered during the idealized calm period. The daily incoming shortwave radiation, $I_o$, was obtained from the Atmospheric Surface Flux Group (ASFG) tower logs at the SHEBA ice station (Persson et al., 2002) (Figure 2). The shortwave radiation records per hour were averaged from July 18th to July 26th, 1998, to estimate the daily variation in the solar variable. For computational efficiency (see below), the averaged data were fitted to a target function $F(t) = A\cos\left(\frac{2\pi t}{B} + \frac{2\pi}{C}\right) + D$, where the constants A, B, C and D were determined by the least-squares method.

The albedos for bare ice and the open sea are 0.63 and 0.06, respectively (Bitz and Lipscomb, 1999; Pegau and Paulson, 2001). Solar radiation is absorbed by the water body and ice cap according to Beer's law with extinction coefficients λ of 0.08 m$^{-1}$ and 1.5 m$^{-1}$, respectively (Jackson et al., 2010; Bettge et al., 1996). An average ice thickness of 2.6 m was obtained from measurements made in the area from July 18th to July 26th (Perovich, et al. 2007). Sea ice sheets melt due to incident solar radiation. The effects of melting on the geometry of sea ice slabs are not considered here; therefore, the study is applied to sea ice sheets large enough and in times short enough to maintain constant sea ice properties (e.g., geometry, mass, salinity).

A nonhydrostatic, Boussinesq, rotating, two-dimensional model is used in this study. The dynamic equations are given by:

$$\frac{\partial u}{\partial t} + u\frac{\partial u}{\partial x} + w\frac{\partial u}{\partial z} = -\frac{1}{\rho_0}\frac{\partial \delta p}{\partial x} + fv + \nu_T \nabla^2 u \qquad (1)$$

$$\frac{\partial v}{\partial t} + u\frac{\partial v}{\partial x} + w\frac{\partial v}{\partial z} = -fu + \nu_T \nabla^2 v \qquad (2)$$

$$\frac{\partial w}{\partial t} + u\frac{\partial w}{\partial x} + w\frac{\partial w}{\partial z} = -\frac{1}{\rho_0}\frac{\partial \delta p}{\partial z} - \frac{1}{\rho_0}\delta\rho g + \nu_T \nabla^2 w \qquad (3)$$

$$\frac{\partial u}{\partial x} + \frac{\partial w}{\partial z} = 0 \qquad (4)$$

for velocity components u, v, w and pressure ($\delta p$) and density ($\delta \rho$) anomalies. At near-frozen temperatures, the density variation is more dependent on salinity than temperature. Thus, the simple equation of state $\rho=0.808\,S+1000$, where S is the salinity in g kg$^{-1}$, is considered (Smith and Morison, 1998). The Coriolis parameter is f=1.4 10$^{-4}$ s$^{-1}$ and g=9.8 m s$^{-2}$. $\nu_T = \nu + \nu_E$, where $\nu=10^{-6}$ m$^2$ s$^{-1}$ is the molecular viscosity and $\nu_E$ is the eddy viscosity coefficient that accounts for the effects of the subgrid scales. The parametrization of $\nu_E$ suggested by Smagorinsky (1963, 1993) is adopted here. This is one of the most popular subgrid scale models parameterizing eddy viscosity. Briefly, the Smagorinsky model assumes that $\nu_E$ is proportional to the absolute strain rate:

$$\nu_E = l_o^2 |S| \tag{5}$$

$$|S| = \sqrt{2 S_{ij} S_{ij}} \tag{6}$$

$$S_{ij} = \frac{1}{2}\left(\frac{\partial u_i}{\partial x_j} + \frac{\partial u_j}{\partial x_i}\right) \tag{7}$$

where $|S|$ is the absolute value of the strain rate tensor S$_{ij}$ and $u_i = \{u, v, w\}$ is the *i-th* velocity component along the $x_i = \{x, y, z\}$ direction. The mixing length scale l$_o$ is parametrized using the Smagorinsky constant Cs, l$_o$=Cs Ls. C$_S$ has been tuned from 0.1 to 0.7, and it is set to 0.2 in the present study, which is a common default value. L$_S$ is a length scale representing a filtering width. Its value depends on the grid discretization. In two-dimensional flows, L$_S$ may be taken as the square root of the area of the computational element. Considering the symmetry of the present setting, the total viscosity $\nu_T$ is then given by:

$$\nu_T = \nu + \underbrace{(C_S\,L_S)^2 \sqrt{\left(\frac{\partial u}{\partial x}\right)^2 + \left(\frac{\partial w}{\partial z}\right)^2 + 0.5\left(\frac{\partial u}{\partial z} + \frac{\partial w}{\partial x}\right)^2 + 0.5\left(\frac{\partial v}{\partial x}\right)^2 + 0.5\left(\frac{\partial v}{\partial z}\right)^2}}_{\nu_E} \tag{8}$$

Free-stress boundary conditions $\frac{\partial u}{\partial z} = 0, \frac{\partial v}{\partial z} = 0, w = 0$ are prescribed at the open water surface and the bottom depth. Nonslip conditions (u=0, v=0, w=0) are specified at the sea ice slabs. Finally, periodic boundary conditions are imposed on the lateral boundaries due to the fringe geometry described above and the periodic repetition pattern expected in the flow under that geometric configuration.

The temperature and salinity fields evolve according to the following equations:

$$\frac{\partial \delta T}{\partial t} + u\frac{\partial \delta T}{\partial x} + w\frac{\partial \delta T}{\partial z} = K_T\,\nabla^2 \delta T + S_r - w\frac{\partial T_{ref}}{\partial z} + K_T\,\frac{\partial^2 T_{ref}}{\partial z^2} \tag{9}$$

$$\frac{\partial \delta S}{\partial t} + u\frac{\partial \delta S}{\partial x} + w\frac{\partial \delta S}{\partial z} = K_S\,\nabla^2 \delta S - w\frac{\partial S_{ref}}{\partial z} + K_S\,\frac{\partial^2 S_{ref}}{\partial z^2} \tag{10}$$

where $\delta T(x, z, t)$ and $\delta S(x, z, t)$ are the temperature and salinity anomaly fields with respect to the reference fields T$_{ref}$(z) and S$_{ref}$(z). A simple thermodynamic model is employed to compute the surface salinity flux at the top interface, Q$_s$. Specifically, Q$_s$ is obtained from:

$$F = \frac{Q_m}{\rho_i L_f} \tag{11}$$

$$Q_s = F(S_o - S_i) \tag{12}$$

where F is the bottom melting rate, $\rho_i$ is a characteristic sea ice density (900 kg m$^{-3}$), L$_f$ is the latent heat of fusion of sea ice (333 kJ kg$^{-1}$), and S$_i$ (4 psu) is the sea ice salinity content. Q$_m$ is the heat flux used to melt the bottom of the ice. Its value is

calculated from the observed bottom melt rates of 0.4 cm d$^{-1}$ in the calm period and 1.2 cm d$^{-1}$ during the storm (Richter-Menge et al. 2001). Melting models (11) and (12) ignore many aspects of sea ice thermodynamics, but they are complex enough to capture the essential features of sea ice melt required in the present idealized framework. Sr is the internal heat source that quantifies the volumetric absorption of solar radiation by the water body (Mao et al., 2010):


$$S_r = H_o\, \lambda\, e^{\lambda z} \tag{13}$$

where $\lambda$ and $H_o$ have been previously defined. $K_{T,S}$ values are defined by (Schumann, 1996):

$\quad K_{T,S} = k_{T,S} + \dfrac{\nu_E}{Pr_{sgs}}$ (14)

where $k_T = 10^{-7}$ m$^2$ s$^{-1}$ ($k_S = 10^{-9}$ m$^2$ s$^{-1}$) is the molecular thermal (salinity) diffusion and $Pr_{sgs} = 0.4$ is the turbulent Prandtl number of the subgrid scale motions (Schumann, 1996).

A Dirichlet boundary condition is prescribed at the bottom of the sea ice slabs ($\delta T(t) = -0.05$ °C) and of the domain ($\delta T(t) = 0.0$

°C). As mentioned before, the latter is justified by the existence of an isothermal layer below the selected domain. A Neumman boundary condition, $\frac{\partial \delta S}{\partial n} = 0$, where n is the direction normal to the boundary, is selected for the salinity field at the open sea surface and the bottom boundary. This boundary condition approximates a zero and negligible salt flux across the open ocean interface and the domain bottom, respectively. Negative salt fluxes define the boundary conditions in the sea ice slabs, representing the flow of fresh water due to sea ice melt. Salinity fluxes across this boundary vary over time, depending on

solar heating. Similar to the velocity field, periodic boundary conditions in $\delta T$ and $\delta S$ are prescribed at the lateral boundaries. The initial conditions are $\delta T(x, z, 0) = 0$ and $\delta S(x, z, 0) = 0$ in the model domain.

The wind simulations are initiated just after the calm period. The intensity of the wind linearly increased for one day up to a nominal value of 6 m s$^{-1}$. This is an idealized numerical representation of the process observed by Richter-Menge et al. (2001) from July 27th to July 31st. A Galilean transformation of the system equations and boundary conditions was performed to

simulate sea ice motion under windy conditions. With this transformation, the coordinate system is held at the sea ice slab while the water domain flows past the lead at constant speed, as in the lead simulations of Kozo (1983), Kantha (1995) and Skyllingstad and Denbo (2001). In combination with the lateral periodic boundary conditions, this numerical configuration describes the motion of the sea ice fringe periodic pattern over the water domain. In addition, it keeps the domain geometry unchanged, avoiding remeshing the domain and speeding up numerical simulations.

The wind stress $\tau$ exerted on the lead open surface is computed according to the following equation:

$$\tau = \rho_a\, Cd\, U_{10}^2 \tag{15}$$

Here, $\rho_a$ (1.2 kg m$^{-3}$ is the density of air, Cd is a dimensionless constant with a value of 1.3×10$^{-3}$ to leading order (Wróbel-

Niedźwiecka et al., 2019), and $U_{10}$ is the wind speed at 10 m high. The balance condition between the shear and wind stresses is:

$$\tau = \rho_o \nu_T\, \frac{\partial u}{\partial z} \tag{16}$$

and is assumed as the surface boundary condition at the open lead.

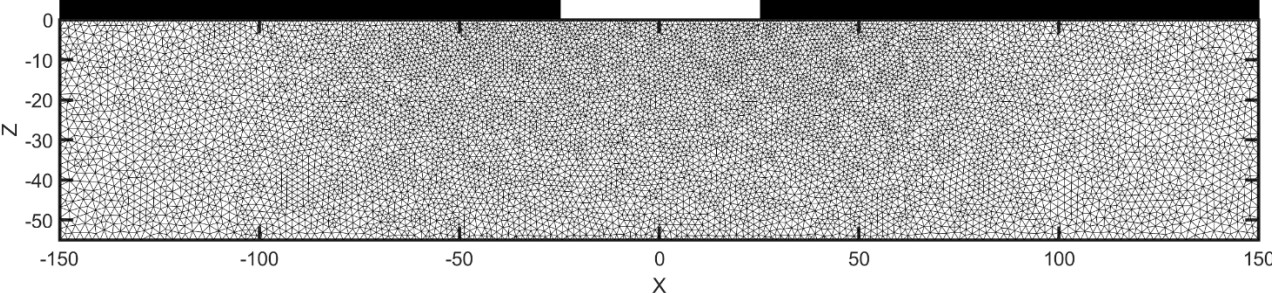

**Figure 3: Triangular meshing of the computational domain. Black rectangles represent location of the sea ice cover.**

## 2.2 Computational approach

A standard Galerkin finite element method has been employed to spatially discretize Eqs. (1-4) and (5-6). The domain geometry was tessellated into 17430 triangular elements with 8952 nodes for this purpose. The characteristic sizes of the elements range from 0.45 m at the open sea surface to 1.8 m at the lateral boundaries (Figure 3). This discretization results in an algebraic system of equations for the values of the fields at the corners of the triangular elements (nodes). Once solved, low-order piecewise polynomials are usually employed to interpolate the solution from the nodes to other locations in the

physical domain. The specific mathematical expressions of the procedure are not replicated here, as they can be found in textbooks about finite elements (e.g., Dhatt and Touzot, 1984; Zienkiewicz and Taylor, 1995).

A fractional step method is selected to integrate in time Eqs. (1-4). The approach is based on operator splitting that yields a decoupling of the convection and diffusion of the velocity and the pressure, which acts to enforce the incompressibility constraint (Chorin, 1968). The time integration of the equations is performed by means of Crank–Nicolson (implicit) time

splitting for the convection and diffusion terms. An adaptive time step is chosen to ensure the stability of the numerical method. The full model was run for five simulation days under calm conditions followed by another five days with wind.

## 3 Results

### 3.1 Calm period

For the analysis of the calm period, the circulation obtained at the end of the five-day simulation is considered. This field constitutes the initial conditions for the following simulations. Figure 4a shows the salinity distribution and current field obtained when the incoming solar radiation (336 W m$^{-2}$) is at its maximum. Sea ice melting creates a layer of relatively fresh water near the surface. There is a horizontal salinity gradient in this layer, with the freshest water below the sea ice slabs rather

than in the lead centre. The horizontal density gradient derived from the salinity variation induces a circulation pattern with almost mirror symmetry with respect to the lead centre (Figure 4a). The latter is described by near-surface convection cells that develop below the lead surface and the sea ice slab up to 4 m deep. The velocity field converges at the geometric centre of the open sea surface and diverges at approximately 2.5 m deep. The maximum velocity in the cells is 0.007 m s$^{-1}$ in this snapshot. The convection cells extend laterally for more than 40 m from the lead centre. A northwards (southwards) flow of

0.005 m s$^{-1}$ develops in the eastern (western) part of the domain.

Convection cells have profound implications for the spatial distribution of the thermal field during this period (Figure 4b). Specifically, surface currents inflowing from the lead sides accumulate warm waters in the lead centre. At this location, warm waters are injected below the sea surface and distributed laterally by convection cells (Figure 4b). The maximum temperature is 1.1 °C at the surface near the centre of the lead. The temperature profile decays exponentially with depth to its reference

value. The thermal drift is the result of heat transport from the surface and heating of subsurface layers in the lead due to solar

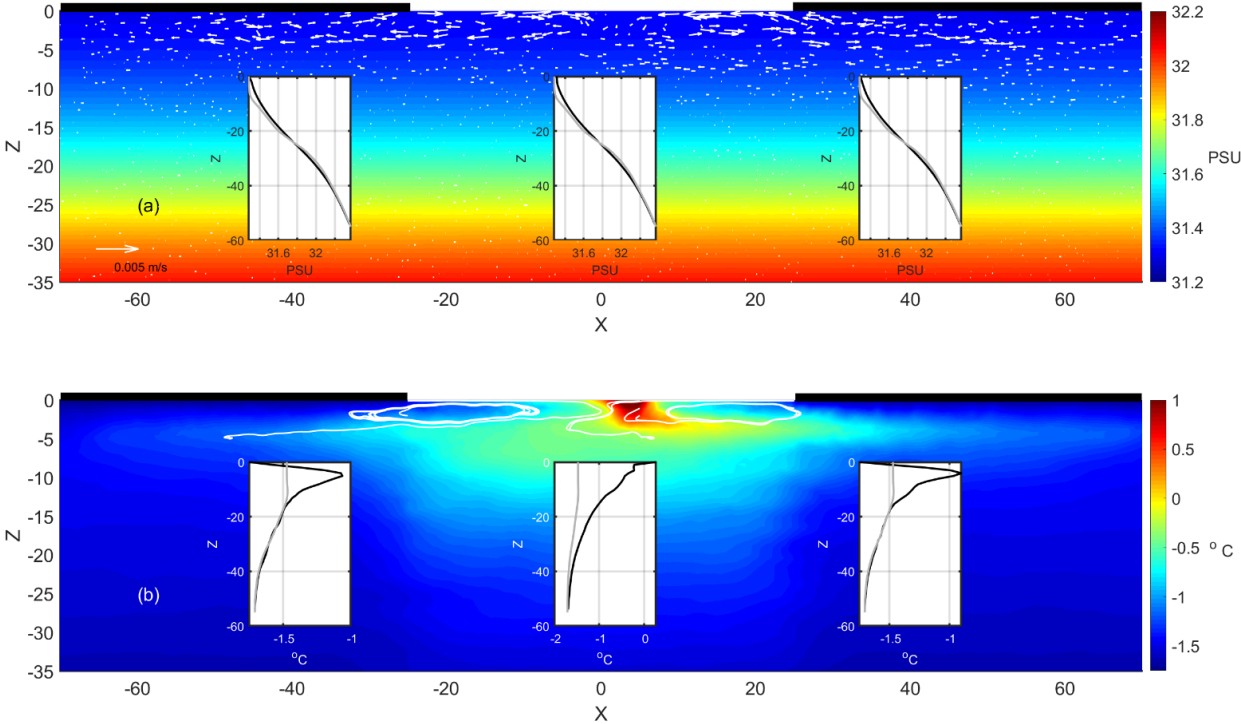

**Figure 4: (a) Salinity and current fields and (b) temperature distribution and streamlines (white lines) at the end of the simulated calm period. The resolution of the current field is reduced to one tenth to facilitate visualization. The inserted plots in panels (a) and (b) compare the salinity and temperature profile (black lines) with the reference profile at the lead center (x=0 m) and at the locations x=-40 m and x=40 m, respectively.**

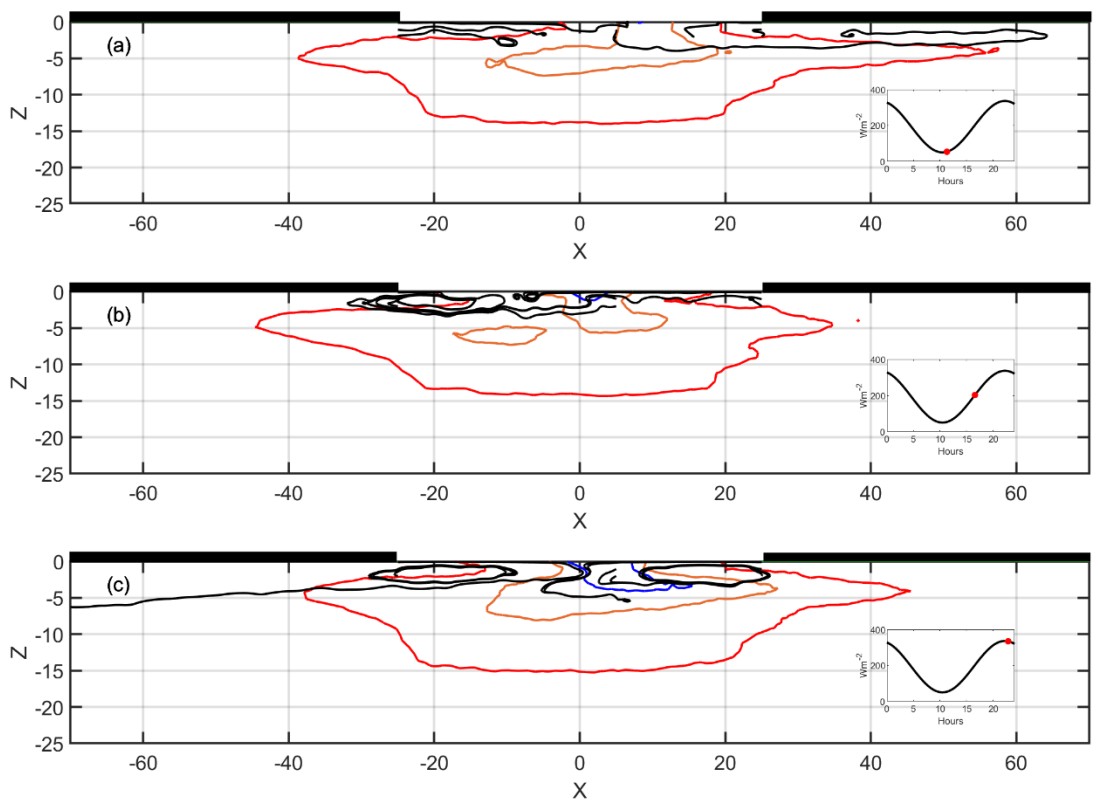

**Figure 5: Isotherms of temperature anomalies of -0.1 ºC (blue), -0.25 ºC (orange) and -0.5 ºC (red) and streamlines during different stages of incoming solar radiation. The inserted plots replicate Figure 2 to facilitate interpretation. The red dot indicates the solar forcing of the corresponding subplot.**

radiation. A local temperature maximum results under the sea ice slabs (Figure 4b). Local maximum temperatures are -0.9 °C and -1 °C at control stations located 40 m east and west of the lead centre, respectively. A net export of heat from the lead to the sea ice slabs results from the convection cells. Heat outflows from the lead region at depths ranging from 3 m to 5 m and partially recirculates between 1 m and 2 m depth due to the velocity pattern induced by the cells. The impact of the cell circulation is limited to the most superficial layers.

The circulation pattern of the cells is not static but undergoes variations due to external forcing and internal dynamics (Figure 5). The cells are not evident at the minimum of solar heating (50 Wm$^{-2}$), existing only in the residual thermal structure of previous heating events (Figure 5a). Cells appear when solar heating increases. Figure 5b shows the initial development of a cell on the western edge of the lead when solar radiation is 190 Wm$^{-2}$. The increase in solar radiation (336 W m$^{-2}$) results in a state with two fully developed cells on both edges of the channel. In addition to this daily cycle, the cell circulation pattern is also modulated by the inertial frequency. Inertial modulations are tracked using the bulk horizontal location of certain isotherms. In particular, the isotherm corresponding to -0.5 °C turned out to be a good indicator of where warm surface waters sink due to the circulation pattern. The bulk horizontal displacement varies mainly with the inertial period (Figure 6). Therefore, the cells alternately stretch and contract due to inertial oscillations.

## 3.2 Wind period

An easterly wind is triggered in the model after the calm period. The intensity of the wind force ramped up during one day until a constant wind speed of 6 ms$^{-1}$ was reached for the remaining days. Sea ice slabs move westwards with the wind at a speed equal to two percent of the wind speed, which is a common wind factor for sea ice cover (Leppäranta, 2011). A westwards flow was generated by the action of wind and ice movement in the resulting final state of the 10-day simulation (Figure 7a). The maximum velocity of the westwards current is 0.15 m s$^{-1}$ at the lead surface and decreases to 0.01 m s$^{-1}$ at 10 m deep. Salinity is homogenized in the water column due to mixing derived from wind and sea ice motion. The mixing effect is more evident at the near-surface temperature (Figure 7b). The snapshot shows a plume at the trailing edge of the moving sea ice slab. The thermal plume is caused by the subduction of warm water under the sea ice due to westwards motion. The heat accumulated by the penetration of solar radiation on the lead surface diffuses rapidly in the upper layers.

A local temperature maximum results at a depth of approximately 28 m as a consequence of the action of the surface stress (Figure 7b). The maximum temperature deepens and reduces its value during the progress of the simulation (Figure 8). The numerical results suggest that the maximum temperature results from the erosion of the initial thermal profile generated during the calm period. The vigorous entrainment, mixing and diffusion of fresh water from melting sea ice driven by currents and turbulent transport removes heat stored during the calm period in the near-surface layers. This generates temperatures in the upper layers that are lower than those in the deeper layers, where the impact of surface stress is weaker and the heat stored during the calm period is thus partially conserved. The warm waters stretch horizontally under the action of surface shear. Thus, an NSTM layer is formed.

## 3.3 Sensitivity studies

### 3.3.1 Buoyancy forcing

The first sensitivity study explores the impact of meltwater runoff on the results described above. The numerical simulations were carried out considering half and double of the reported rate of meltwater runoff, keeping the other parameters the same as in the previous case. This subsection discusses the sensitivity of the NSTM layer to variations in meltwater runoff. The discussion of the results focuses on the impact on the thermal field.

Similar to the case found in Subsection 3.2, the initial warm core stretches horizontally under the action of surface shear when the melt rate is halved (Figure 9a). However, the NSTM layer is warmer and shallower in this scenario. Specifically, the local maximum temperature is now -1.5 °C and is located at a 17 m depth. Figure 9a shows the formation of Kelvin–Helmholtz

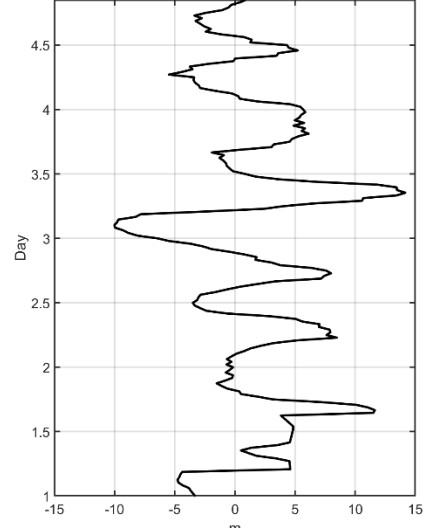

**Figure 6: Time evolution of the bulk horizontal coordinate of the isotherm corresponding to -0.5 ºC.**

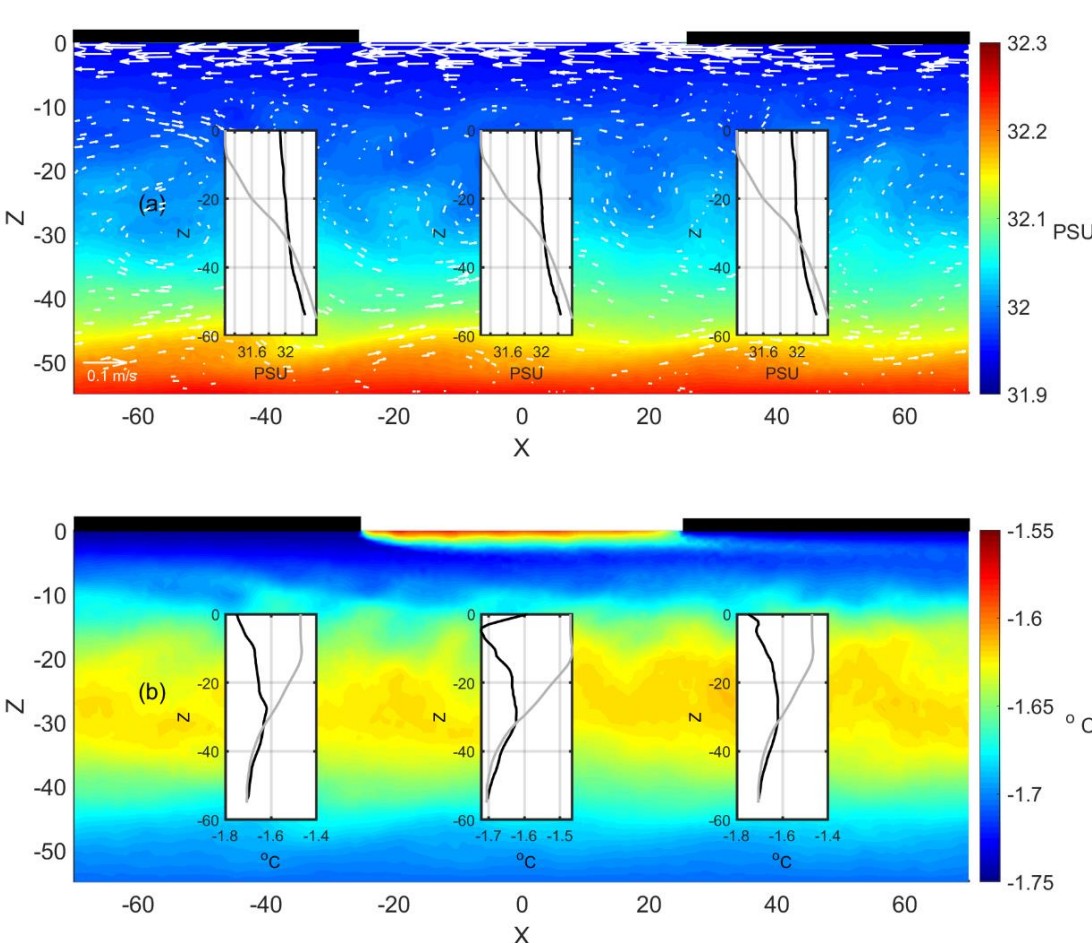

**Figure 7: Similar to Figure 4 but representing the final oceanographic conditions of the water column at the end of the wind period with a wind speed of 6 ms$^{-1}$ and wind factor of 2 %. The Figure represents now the full water column to highlight the near-surface temperature maximum layer.**

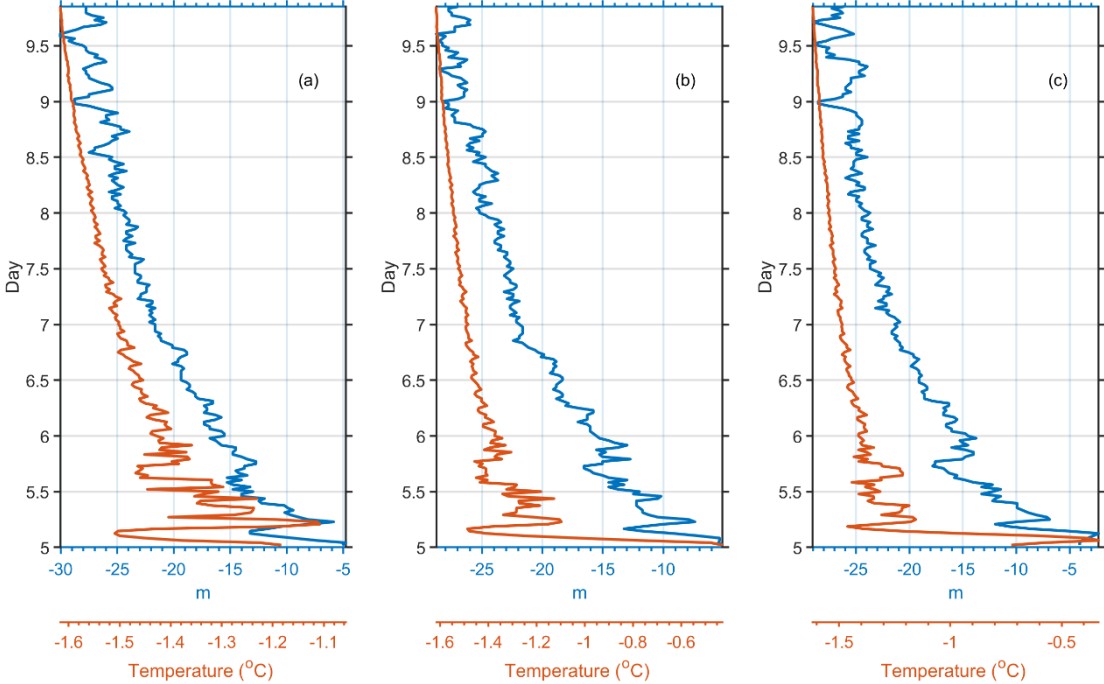

**Figure 8: Evolution of the local maximum temperature (orange line) and its corresponding depth (blue line) in the control stations at (a) x= -40 m, (b) x=0 m and (c) x=40 m. The number on the vertical axis corresponds to the days already completed in the simulation.**

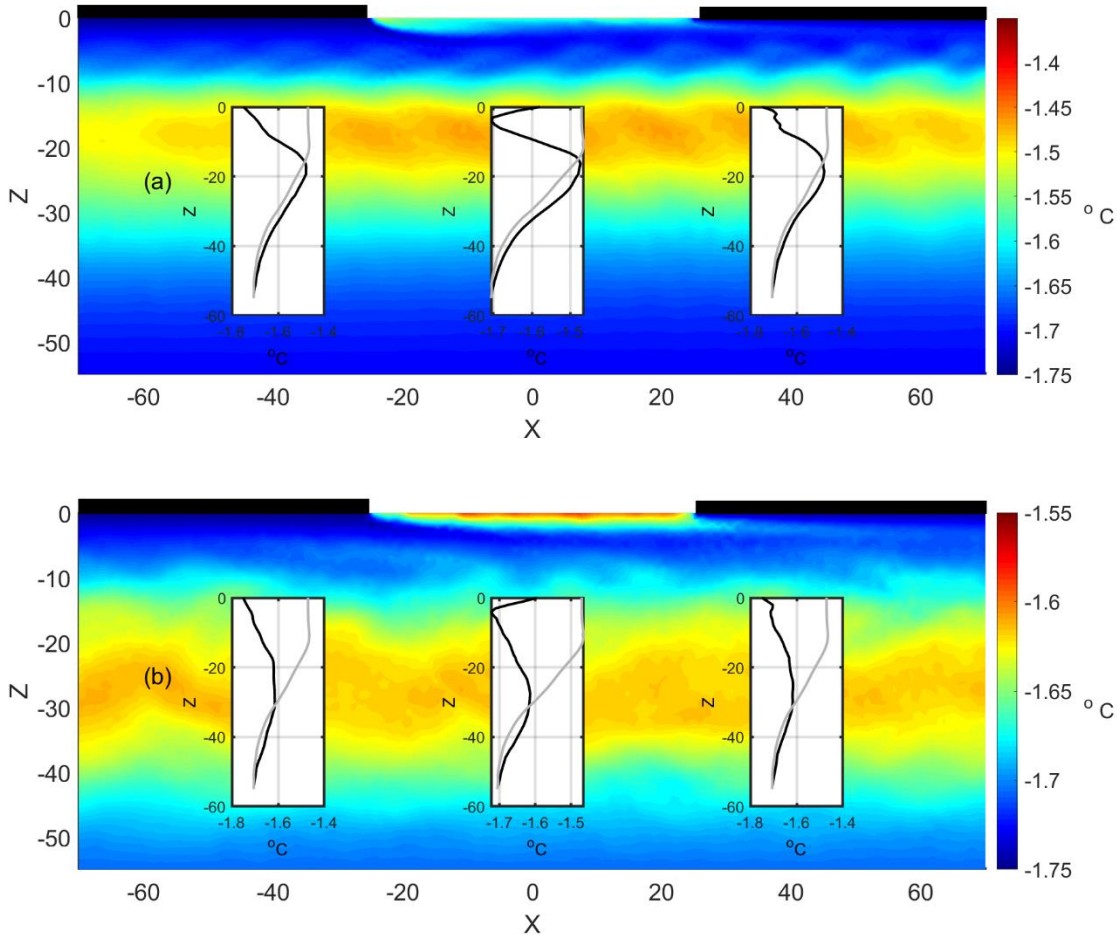

**Figure 9: Similar to Figure 7b but for the case with (a) half and (b) double of the reported melting rate.**

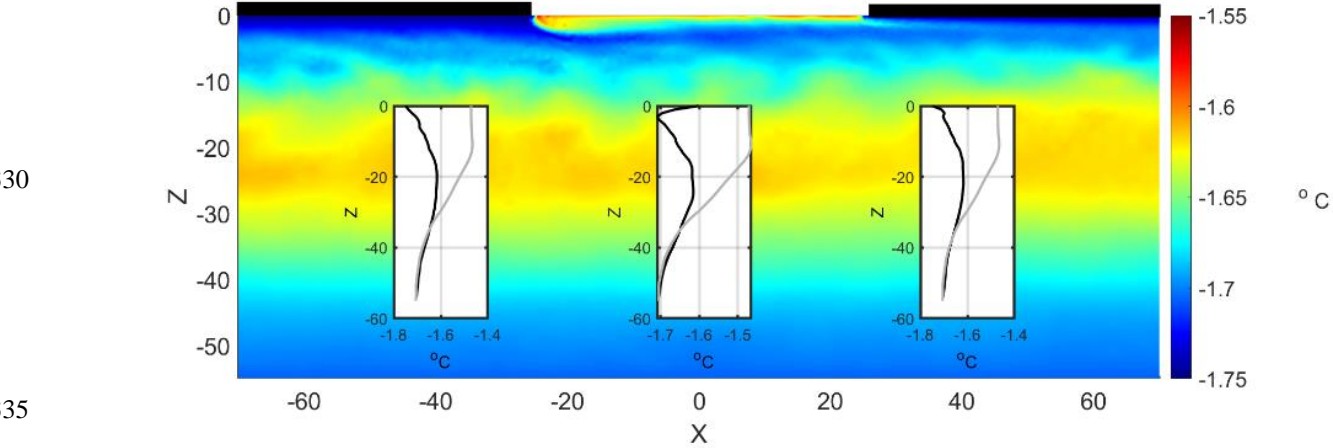

**Figure 10: Similar to Figure 9a but for a model run with 11 days of wind conditions (a total of 16 simulation days including the calm period).**

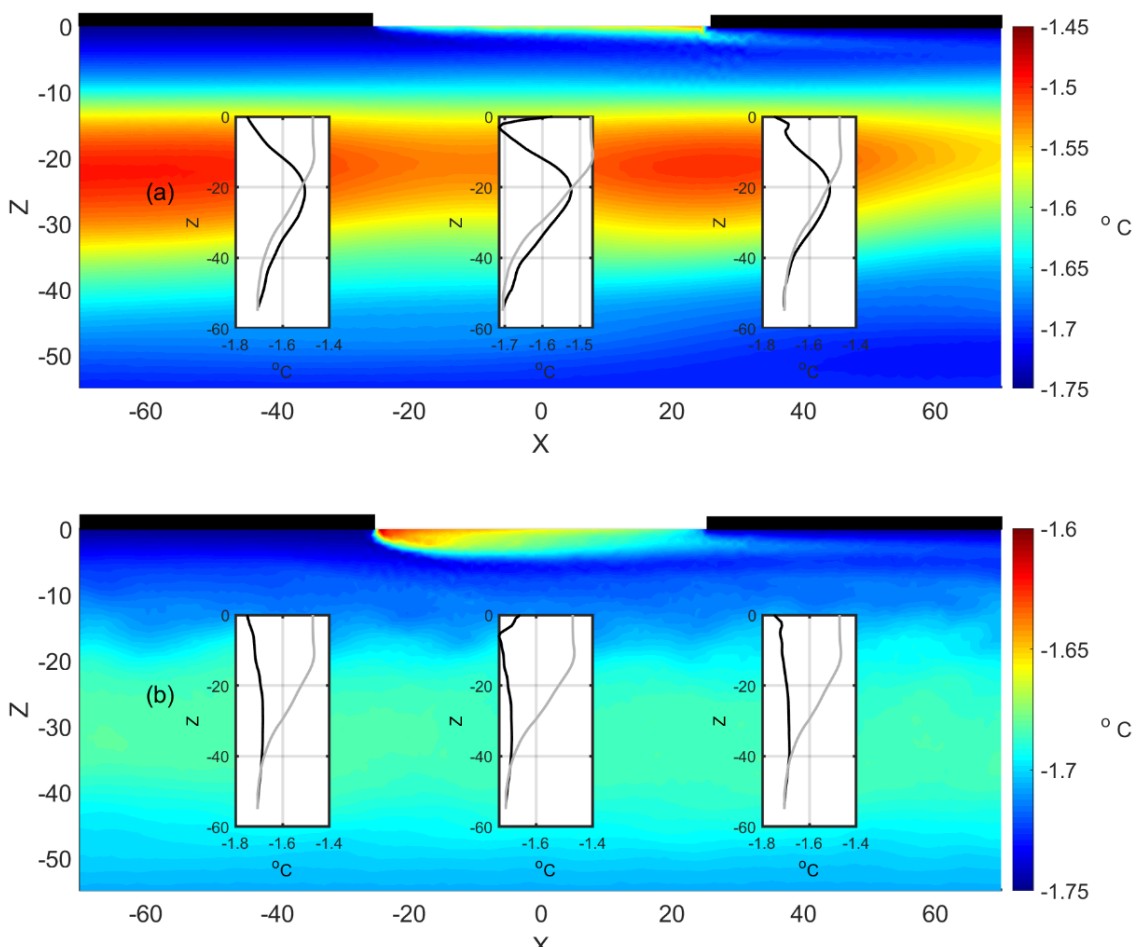

**Figure 11: Thermal distribution at the end of the simulated wind period with wind intensities of (a) 3 ms$^{-1}$ and (b) 9 ms$^{-1}$. The inserted plots in panels (a) and (b) compares the salinity and temperature profile (black lines) with the reference profile at the lead center (x=0 m) and in the control stations at x=-40 m and x=40 m, respectively.**

waves at the interface between the most superficial layers and the NSTM layer. The instability is triggered by the relative motion and density difference between the layers. The bulk Richardson number is 1.2, close to the critical value when the Kelvin–Helmholtz instability is fully developed (Richardson number less than 0.25).

The NSTM layer deepens (28 m depth) and cools down (-1.6 °C) by doubling the initial rate of meltwater runoff (Figure 9b). The pattern and temperature of the NSTM layer closely resembles that of the control run (Figure 7b). The bulk Richardson number is approximately 0.3 for both cases, resulting in a near turbulent interface between the NSTM and overlying layers. A comparison of Figure 9a (run with half melt rate) with Figures 7b and 9b (run with double melt rate) raises a question about the dissimilarity of the former with the other two. The results in Figures 7b and 9b suggest that over a sufficiently long

simulation time, the system reaches an intermediate state characterized by an NSTM layer whose structure does not strongly depend on the rate of meltwater inflow but probably on the range of action in the depth of surface stress. Instead, the inflow rate of the meltwater determines the time required to reach the intermediate state. If this were the case, a state similar to that shown in Figures 7b and 9b would result from a simulation of the case with a halved melt rate subject to a wind period greater than 5 days. Figure 10 shows the result obtained when a wind period of 11 days is applied in the case of a halved melt rate.

The NSTM layer is now less marked than in Figure 9a but shows a more similar structure with the cases shown in Figures 7b and 9b. In particular, the core of the NSTM layer is now below 20 m deep.

### 3.3.2 Wind Forcing

Additional simulations were carried out considering wind intensities of 3 ms$^{-1}$ and 9 ms$^{-1}$. The initialization, wind forcing ramp

up and sea ice motion followed the procedure reported in Subsection 3.2. Under light wind conditions (3 m s$^{-1}$), the NSTM layer is shallower (20 m depth) and warmer (-1.51 °C), as shown in Figure 11a, than in the baseline simulation, as shown in Figure 7b. The original core of warm water under the lead spreads horizontally as it moves west, leaving a trail of warm water at the tail. This generates horizontal thermal anisotropy in the propagation depth of the warm core. Unlike in Figure 7b, no significant development of instabilities is observed along the warm layer.

A local temperature maximum is almost negligible at a 30 m depth when the wind intensity is 9 m s$^{-1}$ (Figure 11b). The thermal profile varies from temperatures close to the freezing point in the surface layers to an almost homogenized thermal background below a 20 m depth. The thermal maximum exceeds the background by only a few hundredths of a degree. Unlike the cases discussed above, the core of slightly warmer waters is now horizontally homogenized. Thermal heating at the lead surface is uncoupled and does not have a significant impact on the NSTM layer during periods of wind and ice motion.


### 3.3.3 Ice motion

As part of the sensitivity studies, a set of numerical simulations evaluated the comparative relevance of the two sources of surface shear stress: wind and sea ice motion. Initially, a period of wind with a wind intensity of 6 ms$^{-1}$ and static sea ice conditions was considered. A second simulation with the same wind intensity but with a wind factor of one percent of the wind

speed complemented the motionless sea ice case in addition to the results displayed in Figure 7b. Again, initialization and wind forcing ramp up followed the procedure detailed in Subsection 3.2. In the static sea ice configuration, the warm core generated under the lead during the calm period is distorted by the surface and near-surface currents induced by the wind stress and the boundary conditions at the fixed edges of the sea ice (Figure 12a). A westwards surface current results in the lead surface by wind action. At the edge of the sea ice, the surface current deepens and bifurcates into a westwards branch beneath

the sea ice and an eastwards recirculation that crosses the lead and penetrates below the eastern sea ice slab. The near-surface current generates shear in the shallower layers (10–15 m depth) of the warm water masses under the lead. Instead, the maximum temperature remains located below the centre of the lead.

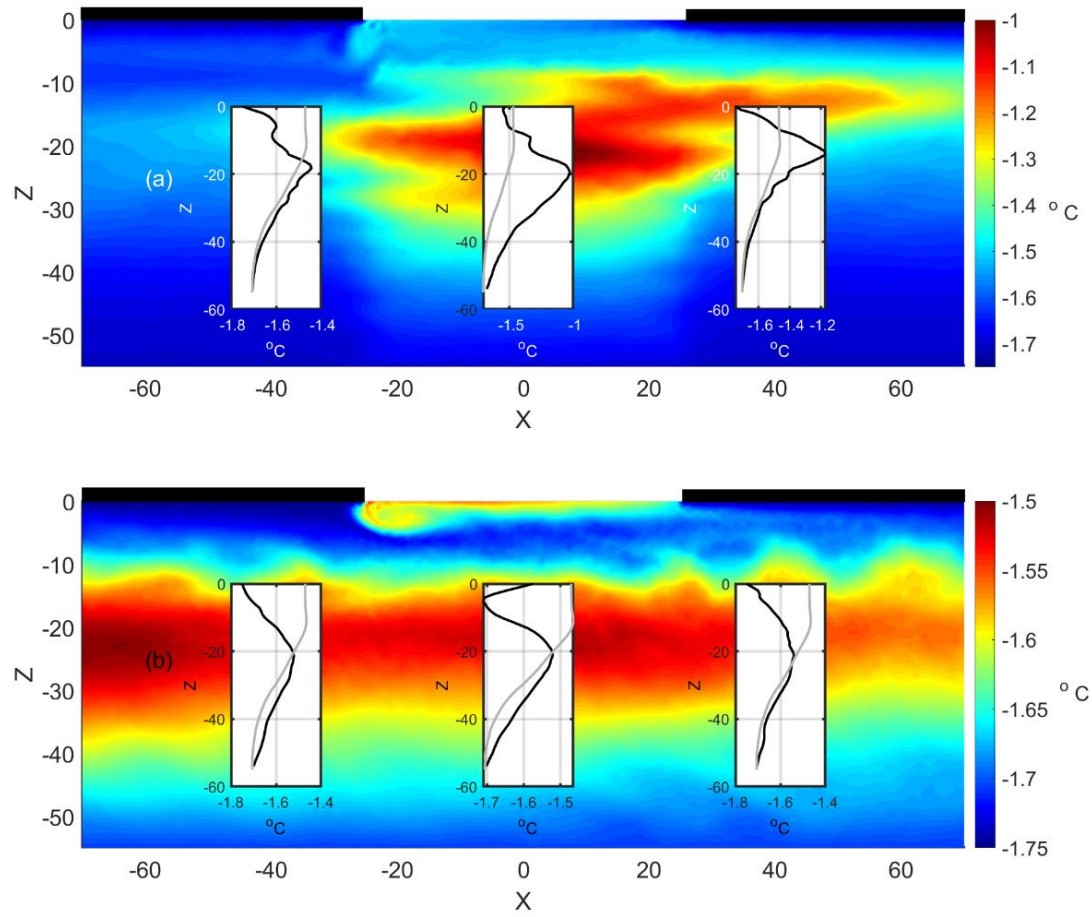

**Figure 12: Thermal distribution at the end of the simulated wind period with wind intensity 6 ms$^{-1}$ and wind factors of (a) 0 % and (b) 1 %. The inserted plots in panels (a) and (b) compares the salinity and temperature profile (black lines) with the reference profile at the lead center (x=0 m) and in the control stations located at x=-40 m and x=40 m, respectively.**

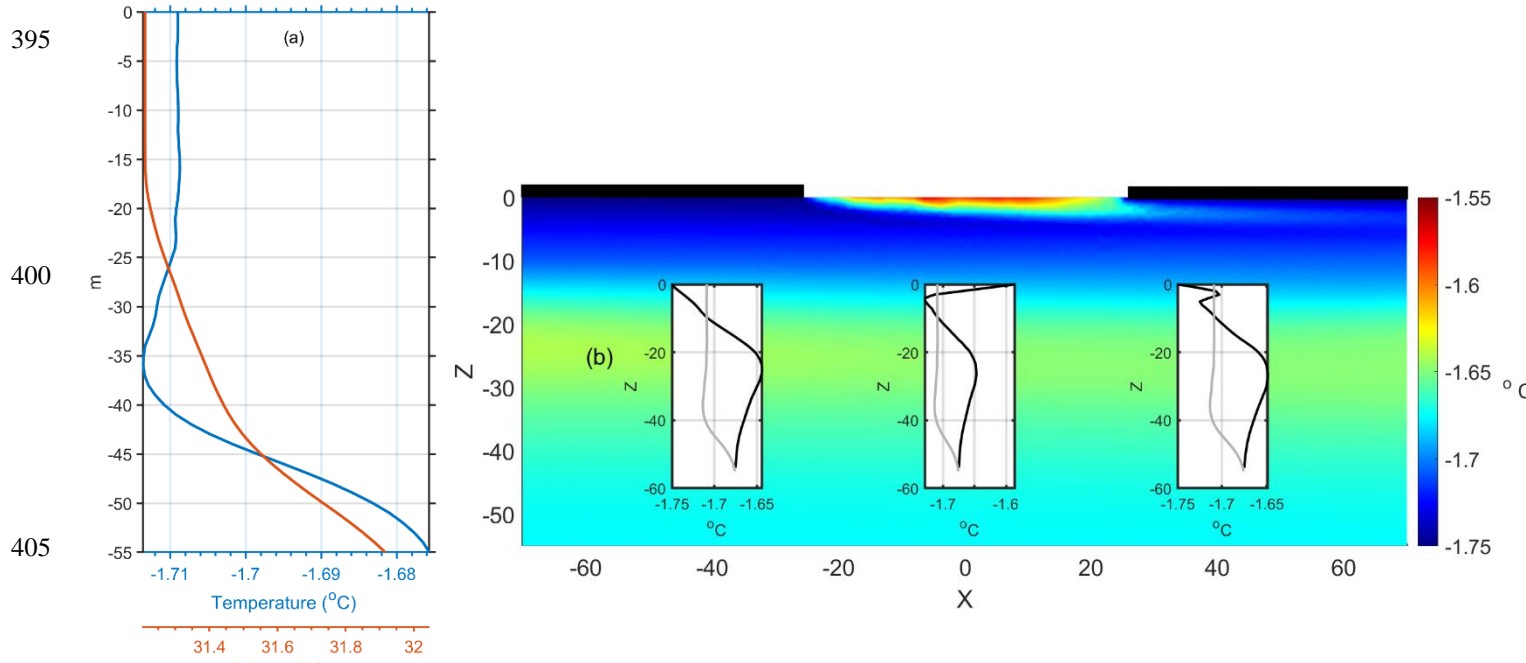

**Figure 13: (a) Reference temperature and salinity profiles and (b) thermal distribution at the end of the simulated wind period with wind intensity 6 ms$^{-1}$ and wind factor of 2%. The inserted plots in panel (b) compare the salinity and temperature profile (black lines) with the reference profile at the lead center (x=0 m) and in the control stations located at x=-40 m and x=40 m, respectively.**

The results differ significantly with sea ice drift (Figure 12b). An NSTM layer is distributed horizontally at a 20 m depth throughout the domain. The temperature distribution is not homogenous in its core, but scattered patches of the original NSTM core are still identified. For this wind factor value, the NSTM layer is warmer and shallower than the case shown in Figure 7b.

3.3.4 Stratification

A different reference stratification of the water column was considered in the numerical study to assess the sensitivity of the reported formation mechanism to this variable. Specifically, Figure 13a displays the stratification conditions considered for this assessment. The reference profiles of temperature and salinity were collected in late April of 2015 at 89.20° N, 55° W in the framework of the project North Pole Station: A Distributed Long-Term Environmental Observatory (Falker and Morison, 2009). Unlike the control stratification (Figure 1), almost homogeneous profiles in temperature and salinity characterize the stratification in the first 35 m depth. In addition to homogeneity, the water column is fresher and colder than the control stratification up to a 45 m depth. Below that depth, the temperature profile is warmer than that observed at the SHEBA location (Figure 1). The numerical simulation followed the same procedure and forcing reported in Subsections 3.1 and 3.2, except for the new reference conditions of stratification.

A local temperature maximum also develops between 20 and 30 m deep for the selected profiles (Figure 13b). The resulting temperature maximum is slightly cooler than that found with the previous stratification (Figure 7b). The NSTM layer clearly differs from the colder upper layer, but the temperature difference between the NSTM and lower layers is less marked. This is a consequence of the fact that the reference temperature profile is warmer at depth than at the surface (contrary to the previously examined thermal profile). The implications of this finding will be discussed below. The spatial variability of the temperature field is smoother than that observed in Figure 7b. This is presumed to be a result of the limited potential energy initially available in the nearly homogeneous density profile (salinity dependent only) down to a 20 m depth to trigger instabilities when surface mixing is active.

*3.3.5 Absence of the calm period*

New numerical simulations evaluate the relevance of the calm period as a conditioning phase for the formation of the NSTM layer. Wind and sea ice movement are activated from the start. Thus, the numerical setup skips the 5 days of calm conditions. In these runs, the model was run for 5 days, as detailed in Subsection 3.2. The profiles displayed in Figures 1 and 13a were considered for initialization.

Figure 14a shows the formation of a layer with a temperature maximum when the stratification displayed in Figure 1 initializes the model. The depth and structure of the emerging NSTM layer resembles that of Figure 7b, where the calm period was considered. In this case, however, the value of the temperature of the NSTM layer is lower than that formed after the calm period. At the NSTM layer depth, the root mean square difference in temperature is approximately 0.02 °C, with local differences up to almost 0.4 °C between both cases. The result can be interpreted in terms of erosion of the relatively warm surface part of the initial profile (Figure 1) by the surface stress. In this circumstance, the heat accumulated during the calm period adds to the initial conditions to increase the value of the temperature maximum.

The profile described in Figure 1 resulted at the SHEBA station after a relatively long period of gentle wind conditions (Ritchter-Menge et al., 2001). Thus, this profile could accumulate the warming resulting from a longer calm period. If true, the relevance of the conditioning phase could be underestimated in the previous experiment. A second model run considered an initial stratification that was not suspected to be subject to previous surface heating, such as the profile analysed in 3.3.4 and shown in Figure 13a. A monotonic increase in temperature with depth is obtained after the 5-day simulation for this profile (Figure 14b). A near surface and localized temperature maximum appears at the trailing edge of the lead. Its dynamic origin is not related to the scope of this study. As in other simulations, it is generated by the advection of surface waters by the wake of


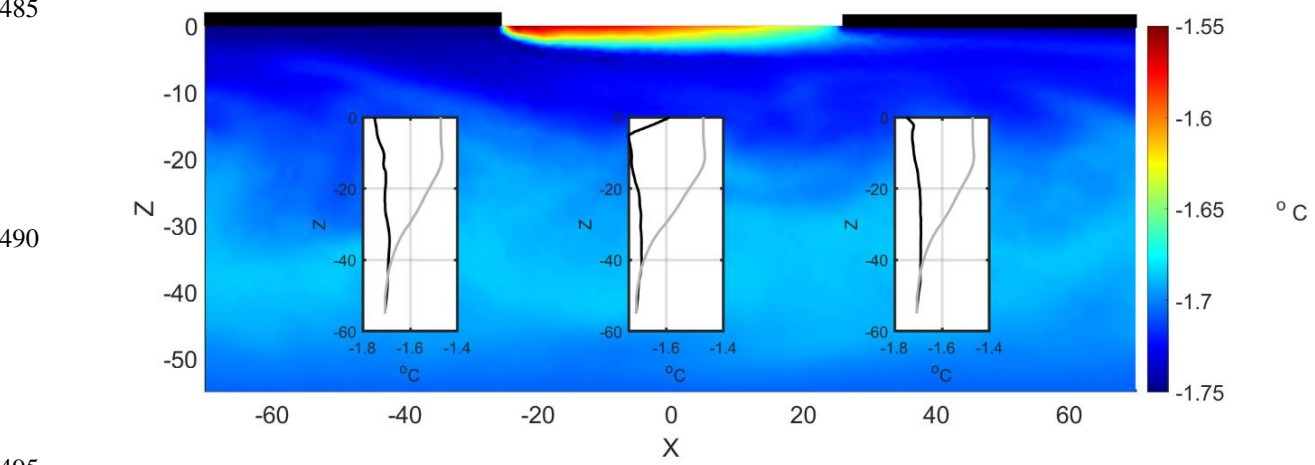





**Figure 14: Thermal distribution at the end of a 5-day simulated wind period with wind intensity 6 ms⁻¹ and wind factor of 2% without considering the calm period. The upper panel (a) shows the results when the run is initialized with the profile shown in Figure 1. The lower panel (b) display the temperature field when the run is initialized with the profile shown in Figure 12a. The inserted plots in panels (a) and (b) compares the salinity and temperature profile (black lines) with the reference profile at the lead center (x=0 m) and in the control stations at x=-40 m and x=40 m, respectively.**





**Figure 15: Thermal distribution at the end of the simulated wind period with wind intensity 6 ms⁻¹ and wind factor of 2% with the new geometry described in Subsection 3.3.6. The inserted plots compare the salinity and temperature profile (black lines) with the reference profile at the lead center (x=0 m) and in the control stations located at x=-40 m and x=40 m, respectively.**

the lead. A comparison between Figures 13b and 14b shows that the existence of a calm period is required for the formation of the NSTM layer in this case.

### 3.3.6 Lead separation

The geometry used in this study is representative of the observed main lead fraction. The separation between leads establishes
the distance between the cores of warm water masses accumulated under the recurring leads during the calm period. The lateral spreading induced by the surface stress connects the remains of the initial warm water cores. The next sensitivity study considers the evolution of the lead system when the distance between leads is almost double (450 m) that in the main geometry considered to date (250 m). The width of the leads remains the same as in previous simulations. The new geometry was meshed using the same procedure as detailed in Subsection 2.2, resulting in 11736 nodes that generate 22839 triangles. The highest
spatial resolution is under the lead, similar to the previous cases. Simulations for the calm and wind periods for the new geometric configuration followed the procedure and forcing reported in Subsections 3.1 and 3.2.

Increasing the lead spacing causes a significant reduction in the temperature of the NSTM layer, which is hardly distinguishable from the background thermal field (Figure 15). This is presumed to be due to the scarce feedback between the cores of warm masses generated in distant leads. Warm water patches extend horizontally for a great distance before connecting with the
remnants of other warm water patches. This increases mixing with the surrounding environment due in part to a larger sea ice cover on the surface. The result is a significant reduction in the thermal signature.

## 4 Discussion

Motivated by observations, this study analysed the relationship between the oceanography below a summer lead exposed to a sequence of calm and moderate wind conditions and the formation of the NSTM layer. Conceptually, the hypothesis assumes that during the calm period, summertime leads are preferential locations to accumulate heat from solar radiation. A subsequent wind event would deepen and horizontally distribute the excess heat accumulated during the calm period under the lead. The horizontal dispersion of heat would be along the direction of wind and sea ice drift. The theoretical framework used to test the
hypothesis considers an idealized scenario that captures the fundamental physical aspects of the problem. These are defined by incoming solar radiation, sea ice melting, lead geometry, winds and ice drift. Other forcings, such as longwave radiation, latent and sensible radiation heat fluxes and surface waves that would exist under real environmental conditions, were not analysed in this study. Paulson and Pegau (2001) reported net average values of -25, 5 and 3 W m$^{-2}$ for the longwave radiation, sensible and latent heat fluxes, respectively, in a lead during summer. The sum of the net radiative, sensible and latent heat
fluxes is small compared with the incoming shortwave radiation and therefore has minimal impact on dynamics at the time scale considered. Finally, wind fetch is quite limited within the leads, preventing the full development of the wave field (Pegau and Paulson, 2001). This is especially true in small leads.

Although the study is carried out in an idealized framework, its design, initial conditions and forcing closely follow the observations made from July 18th to July 31st, 1998, at the SHEBA station. In particular, the conditions during the calm period
are derived from the observations obtained from July 18th to July 26th, while the period from July 27th to July 31st determined certain aspects of the simulations with wind. Unfortunately, the dynamic disparity and the geometric dissimilarity between the idealized framework and the real evolution prevent comparison between model results and observations. In particular, Richter-Menge et al. (2001) reported drastic changes in sea ice field geometry after the storm that are not captured here.

The representation of the sea ice cover in this work follows the same geometrical description found in previous numerical
studies considering leads of similar scales (Kozo, 1983; Smith and Morison, 1993; Smith and Morison, 1998). These are represented by appropriate kinematic and thermodynamic boundary conditions at the sea surface, with no subsurface sea ice structure. Therefore, the proposed idealization excludes the effects of melting at the lateral edge of the sea ice slab and the

effects of the kinematic boundary due to the roughness of the sea ice bottom. The former is negligible for total meltwater runoff due to the small ratio of the leading edge area to the bottom/surface area of the ice cover (Skyllingstad et al., 2005). For the latter, roughness of the sea ice bottom would tend to increase turbulent mixing below the sea ice cover, reducing the difference in density between bottom mixed meltwater runoff and the underlying ocean. Its effect is negligible during the calm period due to the low speeds of the current (less than 0.01 m s$^{-1}$) resulting from the lateral buoyancy gradients between the edges and centre of the lead. Instead, the roughness of the sea ice bottom could be a major factor in the formation of the boundary layer at the bottom of moving sea ice slabs. A detailed quantification of its impact would require a study at finer scales than those considered in this work.

The proposed scenario has been mathematically formalized in terms of a nonhydrostatic, Boussinesq, rotating, two-dimensional model. This mathematical description is common for simulating leads (Kozo, 1983; Smith and Morison, 1998) because it takes full advantage of the symmetry of the lead geometry to reduce the dimensionality of the problem. Its validity is restricted to the lead region away from the boundaries and openings. A detailed simulation of the thermodynamics and dynamics of sea ice is beyond the scope of the present study. Instead, a methodology is required to provide reasonable boundary conditions for salinity fluxes from the melting sea ice cover. In this work, the salinity fluxes resulting from the melting of sea ice have been parametrized based on the observations reported by Richter-Menge et al. 2001. The daily variability in the melt rate assumes a dependence on the incoming solar radiation, while its daily average is equal to the value observed during the calm period. In the absence of observational support, it is assumed that excess melt during the wind period results from basal melting by ocean heat flux. The value of this melting component is constant during the simulated daily cycle.

A daily generation of convection cells near the surface results under the lead and the sea ice slab during the calm period. These result from the combination of lead geometry, solar heating and the melting of sea ice. The convection cells are permanently active during the summer, but their intensity is modulated by the incoming solar radiation cycle. They disappear when solar heating is below a threshold value of approximately 80 W m$^{-2}$. Although relatively restricted to the near-surface layer, convection cells have been revealed as an important mechanism for determining the thermodynamics of the mixed layer under a lead. Convection cells routinely pump heated surface waters below the surface at the lead centre. This causes, together with subsurface heating and heat diffusion, water masses below the lead to be warmer than the surrounding environment. Masses of warm water also extend horizontally below the sea ice cover as a result of the circulation pattern. The patch of masses of warm waters stretches and deepens during the period of maximum solar radiation and contracts when solar heating is minimal.

A net stretching and deepening of warm waters results from the daily cycle process. This new finding explains the deepening and lateral spreading under the adjacent sea ice of warm water masses observed by Richter-Menge et al. (2001) in summertime leads under persistent calm conditions. Furthermore, the model results extend the dynamic analysis to subsurface layers not considered in Richter-Menge et al. (2001). The background conditions of these subsurface layers are a fundamental component in the mechanism leading to the formation of the NSTM layer. Specifically, there is a scientific consensus in attributing the development of the NSTM layer to solar radiation penetrating the upper ocean (Maykut and McPhee, 1995; Jackson et al., 2010, Steele et al., 2011, Gallaher et al., 2017). In this study, we find that subsurface heating in summertime leads is also mediated by the sinking of warm surface waters to the lower layers by convection cells. This mechanism requires buoyancy forcing and lateral boundary conditions at the lead edges, so it does not appear in one-dimensional studies (Gallaher et al., 2017) or in large-scale Arctic simulations where small leads are not resolved (Steele et al., 2011).

A windy period after the calm conditions modifies the previously reported thermal conditions. Shear stress occurs in the uppermost layers from the wind and/or sea ice drift. Regardless of the conditions of sea ice movement, warm water masses become uncoupled from the sea surface. A cold layer of melt runoff insulates the warm water masses from the sea surface, generating an NSTM layer in the numerical results. The NSTM layer develops over ranges of depth and temperature (depending on wind and ice speeds) consistent with observations. In the absence of sea ice drift, the NSTM layer remains localized below the lead. It is dispersed laterally with the motion of the sea ice due to the surface shear stress. The NSTM layer

survives under moderate wind and associated sea ice motion, disappearing for wind intensities greater than 9 m s$^{-1}$. This suggests that the formation of the NSTM layer in leads occurs under a certain range of mixing intensities. A high mixing rate resulting from surface stress would lead either to a homogenization of the water column or to a monotonic dependence of temperature on depth, avoiding the appearance of a temperature maximum.

Numerical simulations suggest that the sequence of calm and windy periods in the leads results in a final thermal structure characterized by a spatially distributed NSTM layer, whose depth and temperature depend on the intensity of the wind and the conditions of sea ice motion. The calm period represents a conditioning phase that reinforces or becomes key to the formation of the NSTM layer, depending on the background thermal stratification. The heat from the incoming solar radiation accumulates below the lead and adjacent areas during this period. The subsequent wind-injected shear stress and sea ice drift

induce turbulent mixing between the warm waters and melt runoff in the upper layers. Mixing reduces the temperature of the superficial layers below the values of the lower layers, the latter being less exposed to the action of surface forcing. Higher forcing and mixing results in a deeper and cooler NSTM layer, as observed in the simulations. The NSTM layer remains detached and isolated from the sea surface by a layer of cooler, fresher water. Despite the different model complexity and resolution, Steele et al. (2011, Figure 2a and corresponding text) found the same mechanism deepening the temperature

maximum layer in their simulations with the Pan-Arctic Ice–Ocean Modelling and Assimilation System (PIOMAS). On the other hand, the hypothesis of the generation of a halocline on top of the NSTM layer that would isolate it from the action of surface forcing (Maykut and McPhee, 1995; Jackson et al., 2010; Gallaher et al., 2017) is not supported here. A halocline separating the upper layer from the NSTM layer does not appear in the numerical simulations. This could be the consequence of the inevitable diffusive character of numerical models. This is known to generate less stratification in the upper layer than

observed (Rosenblum et al., 2021). Alternatively, the reported halocline may develop after the mixing period and be absent during the wind event.

    The two-dimensional physical description of this study highlights other novel aspects related to the formation of the NSTM layer in summer leads. The superficial shear stress generated by the movement of the sea ice dominates that generated by the wind in the horizontal dispersal of subsurface heat. This is due to the limited wind fetch within the leads. In contrast, in a one-

dimensional description of summer lead physics (Gallaher et al., 2017), both sources of surface stress are indistinguishable. This result adds to recent studies highlighting the important role that sea ice surface stress plays in different aspects of Arctic circulation (Dewey et al., 2018).

    The layered structure of the resulting NSTM also emerges from the two-dimensional description used here. Periodic lateral boundary conditions idealize a sea ice landscape made up of large slabs separated by long, narrow open leads, as observed

during the early melting period (Perovich et al. 2001). The surface stress generated by the passage of sea ice slabs laterally spreads the warm waters accumulated under the periodic distribution of leads, connecting the warm waters accumulated on unconnected leads. The location of the NSTM layer in the water column converges to depth values where eroding surface mechanisms (mixing and surface stress) are not effective, with only diffusion remaining as a thermal homogenization process. The latter requires longer time scales than those considered in this study. Another finding of this study relates to the preferential

formation of the NSTM layer in environments where temperature decreases with depth. In contrast, the formation of the NSTM layer is more limited in environments where temperature increases with depth. A monotonic dependence of temperature on depth may result in these scenarios, with the absence of a local temperature maximum. This would occur when the temperature just beneath the cold and fresh mixed layer does not exceed the temperature of the warm layers underneath. This could be the case in Arctic regions where warm Atlantic or Pacific waters flow near the surface.

A significant increase in warming of the upper Arctic Ocean has been observed in recent decades (Steele et al., 2008). During the summer, part of the heat from incoming solar radiation is stored in the NSTM layer formed between mid-June and mid-July (Jackson et al., 2010). Although the formation of the NSTM layer is still under debate, the findings presented here provide a coherent dynamic integration of previous unrelated studies and contribute new elements to the understanding of the formation

of the NSTM in summer leads. While the present results show that the sequence of calm and windy/ice motion periods in summer leads to the formation of an NSTM layer, this local mechanism does not exclude other processes or locations. Steele et al. (2011) reported the generation of the NSTM layer at the basin scale, where large-scale ocean dynamics play a significant role. Therefore, certain combinations of dynamic mechanisms, either on a local or global scale, can result in the formation of NSTMs. Even so, there is a physical picture common to the local and global cases, that is, the heating of the surface layers and a subsequent erosion of the top part of the thermal profile by surface mixing. Some aspects of the formation of the NSTM feature still require further investigation. The role of the halocline during the NSTM formation process is among them.

If predictions are accurate, climate change is expected to lead to a nearly ice-free Arctic Ocean in summer within a few decades. This hypothetical scenario involves the replacement of robust multiyear sea ice by weaker young ice. This would facilitate the formation of melted leads at the beginning of the melting season, increasing their frequency in the ice cover. Therefore, the effects of lead-scale oceanographic processes (including the one reported in this study) may increase their global relevance. A particular motivation for this author is that the expected future warming and deepening of the NSTM layer may result in a sound channel that is similar, but with a different physical origin, to the "Beaufort lens" discovered in the Canadian Arctic (Freitag et al., 2015). Assessing the mechanisms of formation of the NSTM layer, its physical properties, resiliency throughout the year and geographical distribution is of special relevance to determine the underwater soundscape in the future Arctic Ocean, with an impact on the vertical variability of ambient noise and the communication performance of marine mammals.

**Competing interests**

The author declares that he has no conflict of interest.

**Acknowledgments**

This work has been funded by the project SAC000A06/D06 of the NATO Allied Command Transformation-ACT. The author thanks the colleagues in the SHEBA Atmospheric Surface Flux Group, Ed Andreas, Chris Fairall, Peter Guest, and Ola Persson for help collecting and processing the data. The National Science Foundation supported their research with grants to the U.S. Army Cold Regions Research and Engineering Laboratory, NOAA's Environmental Technology Laboratory, and the Naval Postgraduate School. The author thanks two anonymous reviewers for helpful comments and suggestions.

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
