# Peer review of "A model for the Artic mixed layer circulation under a summertime lead: Implications on the near-surface temperature maximum formation"

_The Cryosphere, 2022_

## Author Comment (AC1)

**REVIEWER I**

The manuscript, "A model for the Artic mixed layer circulation under a summertime lead: Implications on the near-surface temperature maximum formation" by Alvarez develops a simplified 2D model of a summertime Arctic lead using SHEBA data to examine the sensitivity of the near surface temperature maximum to changes in solar radiation, winds, and ice motion.

Global climate models have biased salinity and temperature distributions in the upper Arctic Ocean. It has been suggested this could be associated with resolution or parameterizations of vertical mixing (Rosenblum et al. 2020). Future changes in the upper ocean are also very uncertain as the changing sea ice state may modulate ocean circulation. The manuscript touches on an important issue of simulating processes controlling the structure of the upper Arctic Ocean but requires a clearer discussion of the novel conclusions that separate this study from previous studies of the summertime NSTM and to be put in a broader context in terms of the Arctic as a whole and other more complex models. I have listed these concerns below. My recommendation is major revisions.

*I thank the reviewer for his/her comments and suggestions. I detail below the actions taken to answer or clarify them.*

1. Previous studies, such as Richter-Menge et al. (2001), Steele et al. (2011), and Gallaher et al. (2017) have recognized the importance of solar radiation, winds, and sea ice motion in the formation and persistence of the NSTM layer beyond summer. This is mentioned in the discussion, but the results seem to confirm previous findings, not add very much new. Could you expand on the novel results and the benefits of this model within the hierarchy of the models from the other studies?

   *The following paragraphs (in bold) are suggested for inclusion in the Discussion Section to address this comment:*

L 415 ….A net stretching and deepening of warm waters result from the daily cycle process. **This finding of the current work dynamically explains the deepening and lateral spreading under the adjacent ice of warm water masses, observed by Richter-Menge et al., (2001) in summertime leads under persistent calm conditions. Furthermore, model results extend the dynamical analysis to subsurface layers not considered in Richter-Menge et al., (2001). The background conditions of these subsurface layers is a fundamental component in the mechanism for the formation of the NSTM layer. Specifically, there is a scientific consensus in attributing the development of the NSTM layer to the solar radiation penetrating in the upper ocean (Maykut and McPhee, 1995; Jackson et al., 2010, Steele et al., 2011, Gallaher et al., 2017). In this study, we find that subsurface heating in summertime leads is also mediated by the sink of warm surface waters to the lower layers by convection cells. The mechanism develops by the buoyancy forcing and lateral boundary conditions of the lead, so it goes unnoticed in one-dimensional study domains (Gallaher et al., 2017) or in open ocean environments (Steele et al., 2011).**

L 433 ….from the sea surface by a layer of cooler, fresher water. **Despite the different model complexity and resolution, Steele et al., (2011, Figure 2a and corresponding text) found the same mechanism deepening the temperature maximum layer in their simulations with the Pan-Arctic Ice–Ocean Modeling and Assimilation System (PIOMAS). On the other hand, the hypothesis attributing the resilience of the NSTM layer to the protective action of the upper layer on the underneath ones (Maykut and McPhee, 1995; Jackson et al., 2010; Gallaher et al., 2017) is not supported here.**

L438 ….and being absent during the wind event

**The two-dimensional physical description of this study highlights other novel aspects related to the formation of the NSTM layer in summer leads. The superficial shear stress generated by the movement of the ice dominates that generated by the wind in the horizontal dispersal of subsurface heat. This is due to the limited wind fetch within the leads. In contrast, in a one-dimensional description of a summer lead physics (Callaher et al., 2017), both sources of surface stress are indistinguishable. The layered structure of the resulting NSTM also emerges from the two-dimensional description used here. Periodic lateral boundary conditions idealize an ice landscape made up of large plates separated by long, narrow open leads as observed during the early melting period (Perovich et al. 2001). The surface stress generated by the repetitive passage of ice plates spreads laterally the warm waters accumulated under the leads, connecting the warm waters accumulated on unconnected leads. Another finding of this study, relates the preferential formation of the NSTM layer in environments with thermal profiles where the temperature decreases with depth. In the opposite case, eroding the top part of the thermal profile after the calm period may only result on a monotonic increase of temperature with depth. This would occur when the temperature just beneath the cold and fresh mixed layer does not exceed the temperature of the lower warm layers. This could be the case in regions of the Eurasian Arctic if warm Atlantic waters flow near the surface.**

*In addition, the revised manuscript will incorporate new and novel parametric analyses of different aspects of the NSTM formation as a result of this reviewing process. These include new initial stratification (see below), the role of the calm period and the effect of the lead separation.*

2. While there was good discussion about the caveats and limitations of this modeling study, I had a question about a couple more. For example, the model developed here does not include important aspects such as large-scale ocean circulations and only has a very simple representation of freshwater input, which have both been highlighted in the previous studies as important for the NSTM. The model was also developed based on a very short time period from SHEBA and the Arctic has changed considerably since then (Dewey et al., 2018). Does the author expect the conclusions would change with a more recent case? Although it was located in different region of the Arctic, what about with MOSAIC?

*A new sensitivity study using a different and more recent initial profile, would be included to answer this question. In particular, the following section would be incorporated to the study:*

L 361 *3.3.4 Stratification*

**A different initial stratification of the water column was considered in the numerical study to assess the sensitivity of the reported formation mechanism to this variable. Specifically, Figure 12a displays stratification conditions considered for this assessment. The profiles of temperature and salinity were collected in late April of 2015 at 89.20º N, 55º W in the framework of the project North Pole Station: A Distributed Long-Term Environmental Observatory (Kelly and Morison, 2009). Unlike**

[Figure]

**Figure 12: (a) Initial temperature and salinity profiles and (b) thermal distribution at the end of the simulated wind period with wind intensity 6 ms$^{-1}$ and wind factor of 2%. The inserted plots in panels (a) and (b) compares the salinity and temperature profile (black lines) with their reference profiles at the lead center (x=0 m) and in the control stations located at x=-40 m and x=40 m, respectively.**

**the previous case (Figure 1), almost homogeneous profiles in temperature and salinity characterize the stratification conditions in a large part of the range of depths considered. In addition to homogeneity, the water column is fresher and colder than the previous conditions up to 45 m depth. Below that depth, the temperature profile is warmer than the one observed at SHEBA location (Figure 1). The numerical simulation followed the**

same procedure and forcing reported in Subsections 3.1 and 3.2, except for the new initial conditions of stratification.

A local temperature maximum also develops between 20 and 30 m depth for the selected initial profiles (Figure 12b). The resulting temperature maximum is slightly cooler than that found with the previous stratification (Figure 7b). The NSTM layer clearly differs from the colder upper layer, but the temperature difference between the NSTM and lower layers is less marked. This is a consequence of the fact that the initial temperature profile is warmer at depth than at the surface (contrary to the previously examined thermal profile). The implications of this finding will be discussed below. The spatial variability of the temperature field is smoother than that observed in Figure 7b. This is presumed to be a result of the limited potential energy initially available in the nearly homogeneous density profile (salinity dependent only) down to 20 m depth, to trigger instabilities when surface mixing is active.

*These results would be commented in the Discussion Section:*

Another finding of this study, relates the preferential formation of the NSTM layer in environments with thermal profiles where the temperature decreases with depth. In the opposite case, eroding the top part of the thermal profile after the calm period, may only result on a monotonic increase of temperature with depth. This would occur when the temperature just beneath the cold and fresh mixed layer does not exceed the temperature of the lower warm layers. This could be the case in regions of the Eurasian Arctic if warm Atlantic waters flow near the surface.

*The Section will also discuss aspects related to the role of the calm period and the effect of the lead separation.*

3. There are instances throughout the manuscript that require some proofreading. For example, in the paragraph beginning on Line 194, the first and second sentences are missing "are" and "is", respectively and in the last sentence computing should be changed to computing. Another example is in the use of minimum and maximum throughout the manuscript. However, I felt it was well constructed, as I appreciated that the methods section describing the model was easy to follow and the approaches taken are appropriate.

Dewey, S., Morison, J., Kwok, R., Dickinson, S., Morison, D., & Andersen, R. (2018). Arctic ice-ocean coupling and gyre equilibration observed with remote sensing. Geophysical Research Letters, 45(3), 1499-1508.

Rosenblum, E., Fajber, R., Stroeve, J. C., Gille, S. T., Tremblay, L. B., & Carmack, E. C. (2021). Surface salinity under transitioning ice cover in the Canada Basin: Climate model biases linked to vertical distribution of fresh water. Geophysical Research Letters, 48(21), e2021GL094739.

*The revised manuscript will be submitted for professional English editing and proofreading. References will be included in the revised text.*

---

## Author Comment (AC2)

**REVIEWER II**

This is very interesting work, but I think it's missing a few kew elements to make it significant - I am not convinced that it demonstrate the importance of both the calm and windy periods, as is claimed in the abstract and summary.

I thank the reviewer for his/her comments and suggestions. I detail below the actions taken to answer or clarify them.

My main question is about the run setup, where there is a calm period (5 days) before the wind is turned on. Why is that?

*The ultimate goal of this study is to understand the dynamical mechanism of the generation of the NSTM layer in leads as suggested by previous observations (Maykut and McPhee, 1995; Jackson et al., 2010). The main dynamical components would be then the lead geometry, the buoyancy forcing from ice melting and the penetration of solar radiation in the lead. Richter-Menge et al. (2001) provided observations of the shallowest layers in a lead under these conditions (calm conditions). The authors reported a warm layer of fresh water on the surface of the summer leads is the result of the combined effect of meltwater runoff and solar heating. The authors also noticed that this warm freshwater layer deepens and spreads laterally under adjacent ice if calm conditions persist. So the first questions to address was the dynamical origin of this observation and if this deepening of warm waters could be the mechanism for the formation of the NSTM layer. For this reason, simulations during the calm period were first considered. Numerical results highlighted an interesting dynamics during the calm period. However, this dynamical mechanisms alone could not explain the depth at which the NSTM layer is observed. Other dynamical aspects should be included. Steele et al. (2011) and Callaher et al (2017) related in other ocean environments, the deepening of the warm surface waters with surface stress produced by wind or ice movement. The wind simulations in this work investigate if this is the case for leads too. Specifically, the surface stress should be applied after the surface part of the water column has been heated, for example during the calm period. For guidance and data availability, the work of Richter-Menge et al. (2001) nicely fitted for this scope. This fixed the time scales considered in his study.*

*This rationale is summarized in the revised text in:*

L 81 **This study investigates the dynamics in a summer lead exposed to a sequence of calm and moderate wind conditions such as those reported in Richter-Menge et al. (2001) at ice station SHEBA from July 18th to July 31st 1998. An axisymmetric geometry and a particular thermodynamic forcing are common features to summer leads in calm conditions. For this reason, the study initially focuses on the circulation under a summer lead resulting from the combined effect of lead geometry, solar radiation and sea ice melt. Under these conditions, lateral buoyancy gradients between the edges and center of the lead due to solar heating and ice melt can trigger a circulation in the lead. This circulation would result in a warm layer of fresh water on the surface of the lead. The study is then completed with the analysis of the mixing and deepening, if any, of the warm freshwater layer due to wind events (and associated sea ice drift) after the calm period. The ultimate goal of the study is to assess if these environmental conditions could result in the formation of the NSTM layer as inferred from the observations.**

Once the wind is on, even for the smallest wind speed, a 3 m/s wind with a 2% wind factor, would result in a displacement of 2 km per day for the ice. These add up to reasonably large distances over 5 days (particularly for the larger wind experiments), so the periodic nature of the experiment gets a little strange. The ocean field (for the wind forcing cases) is the results of local (calm) initial conditions and a lot of leads that occasionally input more solar radiation in the ocean. Does it make sense to compare only temperature profiles, if we are not sure if the total heat input has been the same?

*The ice landscape during the early melting season has been described as a large recurring plates separated by long, narrow open leads. This ice landscape has been idealized in this work as large (infinite) sequence of ice-patches and open water leads, that is, a periodic fringe pattern with ice patches and open water leads. All leads share the same geometry and are subjected to the same forcing (wind, solar radiation etc) under this symmetry assumption. This mathematical model is complex enough to highlight the aspects we are interested on but simple enough to easily simulate the combined effect of a large set of recurrent leads. Certainly, it is only an idealization of real ice landscapes where the symmetry tries to capture the recurring distribution of leads. The occurrence of leads with different geometry and at different separations is out of the limits of this mathematical description. Furthermore, numerical simulations would be very computationally demanding with the resolutions employed (tens of centimetres) if cyclic symmetry is not exploited.*

*The following clarifications are considered in the revised manuscript*

L 52 … that ice in the leads melts almost completely by the end of July. **At this time, the ice landscape is made up of large recurring plates separated by long, narrow open leads.**

L 94 … **The ice landscape previously described by recurring plates separated by long, narrow open leads, is idealized here by a large (infinite) fringe pattern with ice-patches and open water leads of 250 m and 50 m wide, respectively, Figure 1.**

Maybe I'm missing something… Is the grid (Fig 3) changing as the "Ice sheets are assumed to move westward with the wind"?

*A Galilean transformation is used to hold the coordinate system at the ice plate while the dynamics of the fluid domain is described in a reference frame in motion with respect to the ice. This numerical technique has been used by other authors in the past to simulate leads in motion (Kozo 1983, Kantha 1995 and Skyllingstad and Denbo 2001). The advantage of the approach is that it does not need to modify the mesh of the domain.*

*It is proposed to add the following paragraph to the revised text to clarify this aspect*

L 97 ….. from July 27th to July 31st. **A Galilean transformation of the system equations and boundary conditions was performed to simulate ice motion under windy conditions (Skyllingstad and Denbo, 2001). With this transformation, the coordinate system is held at the ice plate while the water domain flows past the lead at constant speed as in the lead simulations of Kozo (1983), Kantha (1995) and Skyllingstad and Denbo (2001). In combination with the lateral periodic boundary conditions, this numerical configuration describes the motion of the ice fringe periodic pattern over the water domain. In addition, it keeps the domain geometry unchanged, avoiding remeshing the domain and speeding up numerical simulations.**

The results (and modeling setup) make sense to me for the calm period, but it gets confusing when discussing the wind simulation. In particular, the impact of the initial conditions for the 3 m/s case doesn't make sense to me, since this should be the same in all cases - the lead that created the initial warm surface water (Fig 4b) is 10 km away in the 3 m/s case (Fig. 10a), and 20 km away in Fig 7b. Why would only the 3 m/s case show asymmetry (L300)?

*After the calm period, the mathematical model formally describes an infinite sequence of leads with same underwater oceanography due to the imposed periodic symmetry. When the ice moves, the observer is stationary at the top of the ice plate (or lead) represented by the computational domain. As a consequence of the periodic symmetry and the relative motion of the ice, the observer on the ice plate observes the water masses of the leads in the fringe pattern advected past the observer location. Water masses leave the domain at the eastern boundary while the water masses of the next lead enter in the domain through the western boundary. The oceanographic fields evolve during each cycle due to the surface stress generated by the ice motion. In the first cycle, the observed would observe the water masses of the nearest lead perturbed by the pass of one ice stripe, in the second cycle the observed would find the water masses of the second closest lead perturbed by the pass of two ice stripes and so on.*

*The initial cores of water masses spread out horizontally with the number of cycles. This is just a consequence of the surface stress generated by the ice movement. After some time, the initial cores of warm waters are not anymore individually distinguishable but form a more or less homogeneous thermal layer. The number of cycles to observe the layer structure depends on the intensity of the shear stress. The asymmetry in Fig 10a reveals that more than 5 days with wind conditions are required to homogenise the original thermal field for the case of 3 ms^-1.*

It might make more sense to 'close the leads' at the beginning of the wind period, to decouple these two processes (adding more heat or freshwater at the surface), and adding mixing. Ultimately I'm not entirely sure what the simulations demonstrate as they are setup now.

*The numerical simulation suggested by the reviewer was done closing the leads at the beginning of the wind period. The Figure below shows the formation at the end of the 10-day simulation (the usual 5-day with lead and calm conditions plus the 5-day of wind conditions and closed lead). The NSTM layer is cooler than that of Figure 7b due to the increase of shear stress (larger ice surface) and melt water input. Still the simulation shows the resilience of the NSTM feature due to its relative isolation from the surface.*

[Figure]

**Figure: Thermal distribution at the end of a 5-day simulated wind period with wind intensity 6 ms^-1 and wind factor of 2% with closed leads.**

Some of the big questions that this study could really help answer are:

- Do we need calm conditions (ice not moving and lead being open for a long time) to get enough heat in the surface layer to get an NSTM?

[Figure]

**Figure 12a**

*The following results will be included in the revised manuscript to attempt to answer this comment*

**3.3.5 Absence of the calm period**

**New numerical simulations evaluate the relevance of the calm period as a conditioning phase for the formation of the NSTM layer. Wind and ice movement are activated from the start. Thus, the numerical setup skips the 5 days of calm conditions. In these runs, the model was run during 5 days forced as detailed in Subsection 3.2. The profiles displayed in Figures 1 and 12a were considered for initialization.**

**Figure 13a shows the formation of a layer with a temperature maximum when the stratification displayed in Figure 1 initializes the model. The depth and structure of the emerging NSTM layer resembles that of Figure 7b where the calm period was**

[Figure]

**Figure 13: Thermal distribution at the end of a 5-day simulated wind period with wind intensity 6 ms⁻¹ and wind factor of 2% without considering the calm period for the initialization displayed in (a) Figure 1 and (b) Figure 12a. The inserted plots in panels (a) and (b) compares the salinity and temperature profile (black lines) with their reference profiles at the lead center (x=0 m) and in the control stations at x=-40 m and x=40 m, respectively.**

considered. In this case, however, the value of the temperature of the NSTM layer is lower than that formed after the calm period. At the NSTM layer depth, the root mean square difference of temperature is about 0.02ºC with local differences up to almost 0.4 ºC between both cases. The result can be interpreted in terms of erosion of the relatively warm surface part of the initial profile (Figure 1) by the surface stress. In this circumstance, the heat accumulated during the calm period adds to the initial conditions to increase the value of the temperature maximum.

The profile described in Figure 1 resulted at the SHEBA station after a relatively long period of gentle wind conditions (Ritchter-Menge et al., 2001). Thus, this profile could accumulate the warming resulting from a longer calm period. If true, the relevance of the conditioning phase could be underestimated in the previous experiment. A second model run considered an initial stratification that was not suspected to be subject to previous surface heating, such as the profile analyzed in 3.3.4 and shown in Figure 12a. A monotonic increase of temperature with depth is obtained after the 5-day simulation for this profile, Figure 13b. A very shallow and local temperature maximum appears at the trailing edge of the lead. Its dynamical origin is not related with the scope of this study. As in other simulations, it is generated by the advection of surface waters by the wake of the lead. Comparison between Figures 12b and 13b shows that the existence of a calm period is required for the formation of the NSTM layer in this case.

- Do we need (large?) wind events to mix this water to depths comparable to what we observe?

*Calm conditions produce a very localized at the lead's center and near surface (less than 10 m depth) temperature maximum. Numerical results (Figure 8) suggest that mixing by wind or ice movement is required to deepen the temperature maximum to the observed depths (10 to 30 m depth depending of the location). This agrees with the results from Steele et al. (2011) derived for open sea conditions. However, large wind events or moderate winds for long time period could prevent the formation of the NSTM. In the current simulation the thermal signature of the NSTM is almost unnoticed for wind speeds of 9 ms$^{-1}$. That is, the generation and deepening of the NSTM layer seem occurs under certain range of mixing intensity. The following paragraph would be added to the revised manuscript:*

*L 424 …* **The NSTM layer survives under moderate wind and associated ice motion, disappearing for wind intensities greater than 9 ms$^{-1}$. This suggests that the formation of the NSTM layer in leads occurs under certain range of mixing intensity. A high mixing rate resulting from surface stress would lead either to a homogenization of the water column or to a monotonic dependence of temperature with depth, avoiding the appearance of a temperature maximum.**

According to the numerical simulations

- how quickly does the 'patchiness' of the formation (leads) disappear? If one was to increase the separation between leads, would that make the NSTM more patchy?

*The disappearance of the patchiness depends on the intensity of the applied surface stress. For the main case (Figure 7b) the patchiness structure disappears after 3 days of applying the wind forcing and the ice motion. Instead, the patchiness of warm waters*

*would take much longer to disappear if the ice does not move (Figure 11a). Regarding the second question, the following simulation will be added to the revised manuscript:*

**3.3.6 Lead separation**

**The geometry used in this study is representative of the observed main lead fraction. The separation between leads establishes the distance between the cores of warm water masses accumulated under the recurring leads during the calm period. The lateral spreading induced by the surface stress connects the remains of the initial warm water cores. The next sensitivity study considers the evolution of the leads system when the distance between leads is almost double than in the main geometry considered up to now. Specifically, the width of the ice plates separating the leads is 450 m instead the previous width of 250 m. The width of the leads remains the same as in previous simulations. The new geometry was meshed using the same procedure as detailed in sub-Section 2.2, resulting this time in 11736 nodes that generate 22839 triangles. The highest spatial resolution is under the lead similarly to the previous cases. Simulations for the calm and wind period for the new geometric configuration followed the procedure and forcing reported in Subsections 3.1 and 3.2.**

[Figure]

**Figure 13: Thermal distribution at the end of the simulated wind period with wind intensity 6 ms⁻¹ and wind factor of 2% with the new geometry described in Subsection 3.3.6. The inserted plots in panels (a) and (b) compares the salinity and temperature profile (black lines) with their reference profiles at the lead center (x=0 m) and in the control stations located at x=-40 m and x=40 m, respectively.**

**The effect of increasing the separation between leads to a significant reduction of the temperature of the NSTM layer, hardly distinguishable from the background thermal field (Figure 13). This is presumed to be due to the scarce feedback between the cores of warm masses generated in distant leads. Warm water patches extend horizontally for a great distance before connecting with the remnants of other warm water patches. This increases mixing with the surrounding environment due in part to a larger ice sheet on the surface. The result is a significant reduction of the thermal signature.**

I believe that some of the results show here help with these questions, but I would like to see a better justification of the setup.

*The reviewer is referred to the previous answers where this aspect was considered.*

The summary states that "sequence of calm and windy periods in the leads results in a final thermal structure characterized by a spatially distributed NSTM layer" - I'm not sure that this was convincingly demonstrated. Do you *not* get is without the calm periods? A few different scenarios show various levels of patchiness, but the periodic nature of the simulations (when ice is advected) makes it a bit confusing...

*The reviewer is referred to the previous simulations about this subject reported in this document.*

L7: I understand that the focus of the paper is on early summer time, but it might be worth discussing a bit how it connects to winter and seasonality. In particular, while the second sentence of the abstract is probably correct, the heat exchanges are even stronger in the winter. I would probably delete this sentence.

*The sentence will be removed as suggested.*

L229: It is a little bit confusing how how is the 'steady wind period' (L226). From Fig 8, it seems that the wind is turned on at the beginning of "day 5" (the sixth day?)

*The day number is assigned to the completed simulation day, that is, the time period from 0-24 hrs refers to day 1, the period from 24-48 hrs refers to day 2 and so on. The following clarification would be added to the caption of Figure 8:*

**Figure 8….The number on the vertical axis corresponds to the days already completed in the simulation.**

L232: Is Fig. 7 a one-day average, or a snapshot? What does "the circulation on the fifth day is considered" mean? A time average? The next sentence seems to indicate that it is a snapshot in time, but that wasn't clear. Is the circulation (streamlines) calculated from a day-long average?

*It is the final state (snapshot) obtained in the simulation. The following clarifications are proposed:*

**A westward flow has been generated by the action of wind and ice movement in the resulting final state of the 10-day simulation (Figure 7a)**

*in the main text and:*

**Similar to Figure 4 but representing the final oceanographic conditions of the water column at the end of the wind period with a wind speed of 6 ms$^{-1}$ and wind factor of 2 %**

*in the figure caption*

L255: 'Reference profiles' and "Initial profiles" are the same and without lateral variability. They should be called the same (it is a bit confusing here what the gray lines were, and if they changed from panels to panels).

*The notation will be unified as suggested*

L277: It might be useful to add what day "the end of the simulation period" corresponds to (day 10?), to facilitate interpreting Fig 8 in the context of Fig 7 (and others).

*It is suggested to rephrase the sentence:*

**A westward flow has been generated by the action of wind and ice movement in the resulting final state of the 10-day simulation (Figure 7a)**

L 295: The salinity and velocity fields are 'considered' since temperature is effectively a tracer. Rephrase: "Discussion of the results focuses on the impact on the thermal field."

*The sentence will be rephrased in the revised manuscript.*

L300: Specify that the local temperature is warmer, and shallower in this scenario.

*This aspect will be specified in the revised manuscript.*

L304: The 'control run' has also a depth of 28m, and temperatures seems similar (Fig 7b and Fig 9). This should be captured in the text. Ultimately I think that is the main point here - doubling the FW meltrate doesn't impact the NSTM. Why? It seems that the sensitivity is mostly on using smaller melt rates… More melt doesn't change anything (already isolating the temperature maximum), but less has a large impact… Is the depth-integrated heat content the same?

*The observation will be mentioned in the revised manuscript. It is presumed that the states in Figure 7b and 9b are the asymptotic states determine by the depth attenuation of the effects of the surface stress. It would be expected that the case in Figure 9a would approach a similar asymptotic condition for a long enough run. This is because the flow rate of melt waters is smaller and takes longer time to fill the surface layers affected by the surface stress. This hypothesis will be explored and results mentioned in the revised text. The mean heat lost with respect to the final state of the calm simulation is -6.9 $MJ/m^3$, -7.1 $MJ/m^3$ and -7.1 $MJ/m^3$ for cases Figure 9a, Figure 7b and Figure 9b, respectively.*

The last sentence of the abstract states that 'ice drift is key in the development of the NSTM layer'. Closing the lead would do the same, no?

*The reviewer is referred to a previous answer describing the simulation closing the lead during the wind period.*

---

## Author Response (AR1)

**REVIEWER I**

The manuscript, "A model for the Artic mixed layer circulation under a summertime lead: Implications on the near-surface temperature maximum formation" by Alvarez develops a simplified 2D model of a summertime Arctic lead using SHEBA data to examine the sensitivity of the near surface temperature maximum to changes in solar radiation, winds, and ice motion.

Global climate models have biased salinity and temperature distributions in the upper Arctic Ocean. It has been suggested this could be associated with resolution or parameterizations of vertical mixing (Rosenblum et al. 2020). Future changes in the upper ocean are also very uncertain as the changing sea ice state may modulate ocean circulation. The manuscript touches on an important issue of simulating processes controlling the structure of the upper Arctic Ocean but requires a clearer discussion of the novel conclusions that separate this study from previous studies of the summertime NSTM and to be put in a broader context in terms of the Arctic as a whole and other more complex models. I have listed these concerns below. My recommendation is major revisions.

*I thank the reviewer for his/her comments and suggestions. I detail below the actions taken to answer or clarify them.*

1. Previous studies, such as Richter-Menge et al. (2001), Steele et al. (2011), and Gallaher et al. (2017) have recognized the importance of solar radiation, winds, and sea ice motion in the formation and persistence of the NSTM layer beyond summer. This is mentioned in the discussion, but the results seem to confirm previous findings, not add very much new. Could you expand on the novel results and the benefits of this model within the hierarchy of the models from the other studies?

   *The following paragraphs (in bold) is included in the Discussion Section to address this comment:*

L 569 …. A net stretching and deepening of warm waters results from the daily cycle process. **This new finding explains the deepening and lateral spreading under the adjacent ice of warm water masses observed by Richter-Menge et al. (2001) in summertime leads under persistent calm conditions. Furthermore, the model results extend the dynamic analysis to subsurface layers not considered in Richter-Menge et al. (2001). The background conditions of these subsurface layers are a fundamental component in the mechanism leading to the formation of the NSTM layer. Specifically, there is a scientific consensus in attributing the development of the NSTM layer to solar radiation penetrating the upper ocean (Maykut and McPhee, 1995; Jackson et al., 2010, Steele et al., 2011, Gallaher et al., 2017). In this study, we find that subsurface heating in summertime leads is also mediated by the sinking of warm surface waters to the lower layers by convection cells. This mechanism requires buoyancy forcing and lateral boundary conditions at the lead edges, so it does not appear in one-dimensional studies (Gallaher et al., 2017) or in large-scale Arctic simulations where small leads are not resolved (Steele et al., 2011).**

L 598 …. from the sea surface by a layer of cooler, fresher water. **Despite the different model complexity and resolution, Steele et al. (2011, Figure 2a and corresponding text) found the same mechanism deepening the temperature maximum layer in their simulations with the Pan-Arctic Ice–Ocean Modelling and Assimilation System (PIOMAS). On the other hand, the hypothesis of the generation of a halocline on top of the NSTM layer that would isolate it from the action of surface forcing (Maykut and McPhee, 1995; Jackson et al., 2010; Gallaher et al., 2017) is not supported here.**

L606 …. **The two-dimensional physical description of this study highlights other novel aspects related to the formation of the NSTM layer in summer leads. The superficial shear stress generated by the movement of the ice dominates that generated by the wind in the horizontal dispersal of subsurface heat. This is due to the limited wind fetch within the leads. In contrast, in a one-dimensional description of summer lead physics (Gallaher et al., 2017), both sources of surface stress are indistinguishable. This result adds to recent studies highlighting the important role that sea ice surface stress plays in different aspects of Arctic circulation (Dewey et al., 2018).**

**The layered structure of the resulting NSTM also emerges from the two-dimensional description used here. Periodic lateral boundary conditions idealize an ice landscape made up of large slabs separated by long, narrow open leads, as observed during the early melting period (Perovich et al. 2001). The surface stress generated by the passage of sea ice slabs laterally spreads the warm waters accumulated under the periodic distribution of leads, connecting the warm waters accumulated on unconnected leads. Another finding of this study relates to the preferential formation of the NSTM layer in environments where temperature decreases with depth. In contrast, the formation of the NSTM layer is more limited in environments where temperature increases with depth. A monotonic dependence of temperature on depth may result in these scenarios, with the absence of a local temperature maximum. This would occur when the temperature just beneath the cold and fresh mixed layer does not exceed the temperature of the warm layers underneath. This could be the case in Arctic regions where warm Atlantic or Pacific waters flow near the surface.**

*In addition, the revised manuscript incorporates new and novel parametric analyses of different aspects of the NSTM formation as a result of this reviewing process. These include new initial stratification (see below), the role of the calm period and the effect of the lead separation.*

2. While there was good discussion about the caveats and limitations of this modeling study, I had a question about a couple more. For example, the model developed here does not include important aspects such as large-scale ocean circulations and only has a very simple representation of freshwater input, which have both been highlighted in the previous studies as important for the NSTM. The model was also developed based on a very short time period from SHEBA and the Arctic has changed considerably since then (Dewey et al., 2018). Does the author expect the conclusions

would change with a more recent case? Although it was located in different region of the Arctic, what about with MOSAIC?

*A new sensitivity study using a different and more recent reference profile, is included to answer this question. In particular, the following section is incorporated to the study:*

L 417 *3.3.4 Stratification*

**A different reference stratification of the water column was considered in the numerical study to assess the sensitivity of the reported formation mechanism to this variable. Specifically, Figure 13a displays the stratification conditions considered for this assessment. The profiles of temperature and salinity were collected in late April of 2015 at 89.20° N, 55° W in the framework of the project North Pole Station: A Distributed Long-Term Environmental Observatory (Falker and Morison, 2009). Unlike the control stratification (Figure 1), almost homogeneous profiles in temperature and salinity characterize the stratification in the first 35 m depth. In addition to homogeneity, the water column is fresher and colder than the control stratification up to a 45 m depth. Below that depth, the temperature profile is warmer than that observed at the SHEBA location (Figure 1). The numerical simulation followed the same procedure and forcing reported in Subsections 3.1 and 3.2, except for the new initial conditions of stratification.**

**A local temperature maximum also develops between 20 and 30 m deep for the selected profiles (Figure 13b). The resulting temperature maximum is slightly cooler than that found with the previous stratification (Figure 7b). The NSTM layer clearly differs from the colder upper layer, but the temperature difference between the NSTM and lower layers is less marked. This is a consequence of the fact that the initial temperature profile is warmer at depth than at the surface (contrary to the previously examined thermal profile). The implications of this finding will be discussed below. The spatial variability of the temperature field is smoother than that observed in Figure 7b. This is presumed to be a result of the limited potential energy initially available in the nearly homogeneous density profile (salinity dependent only) down to a 20 m depth to trigger instabilities when surface mixing is active.**

[Figure]

**Figure 13: (a) Initial temperature and salinity profiles and (b) thermal distribution at the end of the simulated wind period with wind intensity 6 ms⁻¹ and wind factor of 2%. The inserted plots in panels (a) and (b) compares the salinity and temperature profile (black lines) with their reference profiles at the lead center (x=0 m) and in the control stations located at x=-40 m and x=40 m, respectively.**

*These results are commented in the Discussion Section:*

L616. **Another finding of this study relates to the preferential formation of the NSTM layer in environments where temperature decreases with depth. In contrast, the formation of the NSTM layer is more limited in environments where temperature increases with depth. A monotonic dependence of temperature on depth may result in these scenarios, with the absence of a local temperature maximum. This would occur when the temperature just beneath the cold and fresh mixed layer does not exceed the temperature of the warm layers underneath. This could be the case in Arctic regions where warm Atlantic or Pacific waters flow near the surface.**

*The Section also discuss aspects related to the role of the calm period and the effect of the lead separation.*

3. There are instances throughout the manuscript that require some proofreading. For example, in the paragraph beginning on Line 194, the first and second sentences are missing "are" and "is", respectively and in the last sentence computing should be changed to computing. Another example is in the use of minimum and maximum throughout the manuscript. However, I felt it was well constructed, as I appreciated that the methods section describing the model was easy to follow and the approaches taken are appropriate.

Dewey, S., Morison, J., Kwok, R., Dickinson, S., Morison, D., & Andersen, R. (2018). Arctic ice-ocean coupling and gyre equilibration observed with remote sensing. Geophysical Research Letters, 45(3), 1499-1508.

Rosenblum, E., Fajber, R., Stroeve, J. C., Gille, S. T., Tremblay, L. B., & Carmack, E. C. (2021). Surface salinity under transitioning ice cover in the Canada Basin: Climate model biases linked to vertical distribution of fresh water. Geophysical Research Letters, 48(21), e2021GL094739.

*The revised manuscript was submitted for professional English editing and proofreading. References are included in the revised text.*

**REVIEWER II**

This is very interesting work, but I think it's missing a few kew elements to make it significant - I am not convinced that it demonstrate the importance of both the calm and windy periods, as is claimed in the abstract and summary.

I thank the reviewer for his/her comments and suggestions. I detail below the actions taken to answer or clarify them.

My main question is about the run setup, where there is a calm period (5 days) before the wind is turned on. Why is that?

*The ultimate goal of this study is to understand the dynamical mechanism of the generation of the NSTM layer in leads as suggested by previous observations (Maykut and McPhee, 1995; Jackson et al., 2010). The main dynamical components would be then the lead geometry, the buoyancy forcing from ice melting and the penetration of solar radiation in the lead. Richter-Menge et al. (2001) provided observations of the shallowest layers in a lead under these conditions (calm conditions). The authors reported a warm layer of fresh water on the surface of the summer leads is the result of the combined effect of meltwater runoff and solar heating. The authors also noticed that this warm freshwater layer deepens and spreads laterally under adjacent ice if calm conditions persist. So the first questions to address was the dynamical origin of this observation and if this deepening of warm waters could be the mechanism for the formation of the NSTM layer. For this reason, simulations during the calm period were first considered. Numerical results highlighted an interesting dynamics during the calm period. However, this dynamical mechanisms alone could not explain the depth at which the NSTM layer is observed. Other dynamical aspects should be included. Steele et al. (2011) and Gallaher et al (2017) related in other ocean environments, the deepening of the warm surface waters with surface stress produced by wind or ice movement. The wind simulations in this work investigate if this is the case for leads too. Specifically, the surface stress should be applied after the surface part of the water column has been heated, for example during the calm period. For guidance and data availability, the work of Richter-Menge et al. (2001) nicely fitted for this scope. This fixed the time scales considered in his study.*

*This rationale is summarized in the revised text in:*

L 82 **This study investigates the dynamics in a summer lead exposed to a sequence of calm and moderate wind conditions such as those reported in Richter-Menge et al. (2001) at the SHEBA ice station from July 18th to July 31st, 1998. An axisymmetric geometry and a particular thermodynamic forcing are common features of summer leads under calm conditions. For this reason, the study initially focuses on the circulation under a summer lead resulting from the combined effects of lead geometry, solar radiation and sea ice melt. Under these conditions, lateral buoyancy gradients between the edges and centre of the lead due to solar heating and ice melt can trigger circulation in the lead. This circulation would result in a warm layer of fresh water on the surface of the lead. The study is then completed with an analysis of the mixing and deepening, if any, of the warm freshwater layer due to wind events (and associated sea ice drift) after the calm period. The ultimate goal of the study is to assess whether these environmental conditions from field observations could result in the formation of an NSTM layer as inferred from the observations.**

Once the wind is on, even for the smallest wind speed, a 3 m/s wind with a 2% wind factor, would result in a displacement of 2 km per day for the ice. These add up to reasonably large distances over 5 days (particularly for the larger wind experiments), so

the periodic nature of the experiment gets a little strange. The ocean field (for the wind forcing cases) is the results of local (calm) initial conditions and a lot of leads that occasionally input more solar radiation in the ocean. Does it make sense to compare only temperature profiles, if we are not sure if the total heat input has been the same?

*The ice landscape during the early melting season has been described as a large recurring sea ice slabs separated by long, narrow open leads. This sea ice landscape has been idealized in this work as large (infinite) sequence of ice-patches and open water leads, that is, a periodic fringe pattern with ice slabs and open water leads. All leads share the same geometry and are subjected to the same forcing (wind, solar radiation etc) under this symmetry assumption. This mathematical model is complex enough to highlight the aspects we are interested on but simple enough to easily simulate the combined effect of a large set of recurrent leads. Certainly, it is only an idealization of real ice landscapes where the symmetry tries to capture the recurring distribution of leads. The occurrence of leads with different geometry and at different separations is out of the limits of this mathematical description. Furthermore, numerical simulations would be very computationally demanding with the resolutions employed (tens of centimetres) if cyclic symmetry is not considered.*

*The following clarifications are included in the revised manuscript*

L 53 … that ice in the leads melts almost completely by the end of July. **At this time, the ice landscape is made up of large recurring sea ice slabs separated by long, narrow open leads.**

L 96 … **The landscape was idealized here by a large (infinite) fringe pattern with sea ice slabs and open water leads 250 m and 50 m wide, respectively (Figure 1)**

Maybe I'm missing something… Is the grid (Fig 3) changing as the "Ice sheets are assumed to move westward with the wind"?

*A Galilean transformation is used to hold the coordinate system at the sea ice slab while the dynamics of the fluid domain is described in a reference frame in motion with respect to the sea ice. This numerical technique has been used by other authors in the past to simulate leads in motion (Kozo 1983, Kantha 1995 and Skyllingstad and Denbo 2001). The advantage of the approach is that it does not need to modify the mesh of the domain.*

*The following paragraph is added to the revised text to clarify this aspect*

L198 ….. **A Galilean transformation of the system equations and boundary conditions was performed to simulate ice motion under windy conditions. With this transformation, the coordinate system is held at the sea ice slab while the water domain flows past the lead at constant speed, as in the lead simulations of Kozo (1983), Kantha (1995) and Skyllingstad and Denbo (2001). In combination with the lateral periodic boundary conditions, this numerical configuration describes the motion of the sea ice fringe periodic pattern over the water domain. In addition, it keeps the domain geometry unchanged, avoiding remeshing the domain and speeding up numerical simulations.**

The results (and modeling setup) make sense to me for the calm period, but it gets confusing when discussing the wind simulation. In particular, the impact of the initial conditions for the 3 m/s case doesn't make sense to me, since this should be the same in all cases - the lead that created the initial warm surface water (Fig 4b) is 10 km away in the 3 m/s case (Fig. 10a), and 20 km away in Fig 7b. Why would only the 3 m/s case show asymmetry (L300)?

*After the calm period, the mathematical model formally describes an infinite sequence of leads with same underwater oceanography due to the imposed periodic symmetry. When the ice moves, the observer is stationary at the top of the sea ice slab (or lead) represented by the computational domain. As a consequence of the periodic symmetry and the relative motion of the ice, the observer on the ice plate observes the water masses of the leads in the fringe pattern advected past the observer location. Water masses leave the domain at the eastern boundary while the water masses of the next lead enter in the domain through the western boundary. The oceanographic fields evolve during each cycle due to the surface stress generated by the ice motion. In the first cycle, the observed would observe the water masses of the nearest lead perturbed by the pass of one sea ice slab, in the second cycle the observed would find the water masses of the second closest lead perturbed by the pass of two sea ice slabs and so on.*

*The initial cores of water masses spread out horizontally with the number of cycles. This is just a consequence of the surface stress generated by the ice movement. After some time, the initial cores of warm waters are not anymore individually distinguishable but form a more or less homogeneous thermal layer. The number of cycles to observe the layer structure depends on the intensity of the shear stress. The asymmetry in Fig 10a reveals that more than 5 days with wind conditions are required to homogenise the original thermal field for the case of 3 ms$^{-1}$.*

It might make more sense to 'close the leads' at the beginning of the wind period, to decouple these two processes (adding more heat or freshwater at the surface), and adding mixing. Ultimately I'm not entirely sure what the simulations demonstrate as they are setup now.

*The numerical simulation suggested by the reviewer was done closing the leads at the beginning of the wind period. The Figure below shows the formation at the end of the 10-day simulation (the usual 5-day with lead and calm conditions plus the 5-day of wind conditions and closed lead). The NSTM layer is cooler than that of Figure 7b due to the increase of shear stress (larger ice surface) and melt water input. Still the simulation shows the resilience of the NSTM feature due to its relative isolation from the surface. The depths of the NSTM layer are not impacted by the surface heating at the lead during the wind simulations. The sea ice-lead-sea ice geometry remains constant during the motion because the sea ice slabs move at the same speed.*

[Figure]

**Figure: Thermal distribution at the end of a 5-day simulated wind period with wind intensity 6 ms$^{-1}$ and wind factor of 2% with closed leads.**

Some of the big questions that this study could really help answer are:

- Do we need calm conditions (ice not moving and lead being open for a long time) to get enough heat in the surface layer to get an NSTM?

*The following results is included in the revised manuscript to attempt to answer this comment*

**L 436 3.3.5 Absence of the calm period**

**New numerical simulations evaluate the relevance of the calm period as a conditioning phase for the formation of the NSTM layer. Wind and ice movement are activated from the start. Thus, the numerical setup skips the 5 days of calm conditions. In these runs, the model was run for 5 days, as detailed in Subsection 3.2. The profiles displayed in Figures 1 and 13a were considered for initialization.**

**Figure 14a shows the formation of a layer with a temperature maximum when the stratification displayed in Figure 1 initializes the model. The depth and structure of the emerging NSTM layer resembles that of Figure 7b, where the calm period was considered. In this case, however, the value of the temperature of the NSTM layer is lower than that formed after the calm period. At the NSTM layer depth, the root mean square difference in temperature is approximately 0.02 °C, with local differences up to almost 0.4 °C between both cases. The result can be interpreted in terms of erosion of the relatively warm surface part of the initial profile (Figure 1) by the surface stress. In this circumstance, the heat accumulated during the calm period adds to the initial conditions to increase the value of the temperature maximum.**

**The profile described in Figure 1 resulted at the SHEBA station after a relatively long period of gentle wind conditions (Ritchter-Menge et al., 2001). Thus, this profile could accumulate the warming resulting from a longer calm period. If true, the relevance of the conditioning phase could be underestimated in the previous experiment. A second model run considered an initial stratification that was not suspected to be subject to previous surface heating, such as the profile analysed in 3.3.4 and shown in Figure 13a. A monotonic increase in temperature with depth is obtained after the 5-day simulation for this profile (Figure 14b). A near surface and localized temperature maximum appears at the trailing edge of the lead. Its dynamic origin is not related to the scope of this study. As in other simulations, it is generated by the advection of surface waters by the wake of the lead. A comparison between Figures 13b and 14b shows that the existence of a calm period is required for the formation of the NSTM layer in this case.**

**Figure 13: (a) Initial temperature and salinity profiles**

[Figure]

**Figure 14: Thermal distribution at the end of a 5-day simulated wind period with wind intensity 6 ms⁻¹ and wind factor of 2% without considering the calm period for the initialization displayed in (a) Figure 1 and (b) Figure 12a. The inserted plots in panels (a) and (b) compares the salinity and temperature profile (black lines) with their reference profiles at the lead center (x=0 m) and in the control stations at x=-40 m and x=40 m, respectively.**

- Do we need (large?) wind events to mix this water to depths comparable to what we observe?

*Calm conditions produce a very localized at the lead's center and near surface (less than 10 m depth) temperature maximum. Numerical results (Figure 8) suggest that mixing by wind or ice movement is required to deepen the temperature maximum to the observed depths (10 to 30 m depth depending of the location). This agrees with the results from Steele et al. (2011) derived for open sea conditions. However, large wind events or moderate winds for long time period could prevent the formation of the NSTM. In the current simulation the thermal signature of the NSTM is almost unnoticed for wind speeds of 9 ms⁻¹(Figure 11b). That is, the generation and deepening of the NSTM layer seem occurs under certain range of mixing intensity. The following paragraph would be added to the revised manuscript:*

*L 584 …* **The NSTM layer survives under moderate wind and associated ice motion, disappearing for wind intensities greater than 9 m s⁻¹. This suggests that the formation of the NSTM layer in leads occurs under a certain range of mixing intensities. A high mixing rate resulting from surface stress would lead**

either to a homogenization of the water column or to a monotonic dependence of temperature on depth, avoiding the appearance of a temperature maximum.

According to the numerical simulations

- how quickly does the 'patchiness' of the formation (leads) disappear? If one was to increase the separation between leads, would that make the NSTM more patchy?

*The disappearance of the patchiness depends on the intensity of the applied surface stress. For the main case (Figure 7b) the patchiness structure disappears after 3 days of applying the wind forcing and the ice motion. Instead, the patchiness of warm waters would take much longer to disappear if the ice does not move (Figure 12a). Regarding the second question, the following simulation will be added to the revised manuscript:*

**L 502** *3.3.6 Lead separation*

**The geometry used in this study is representative of the observed main lead fraction. The separation between leads establishes the distance between the cores of warm water masses accumulated under the recurring leads during the calm period. The lateral spreading induced by the surface stress connects the remains of the initial warm water cores. The next sensitivity study considers the evolution of the lead system when the distance between leads is almost double (450 m) that in the main geometry considered to date (250 m). The width of the leads remains the same as in previous simulations. The new geometry was meshed using the same procedure as detailed in Subsection 2.2, resulting in 11736 nodes that generate 22839 triangles. The highest spatial resolution is under the lead, similar to the previous cases. Simulations for the calm and wind periods for the new geometric configuration followed the procedure and forcing reported in Subsections 3.1 and 3.2.**

[Figure]

**Figure 15: Thermal distribution at the end of the simulated wind period with wind intensity 6 ms$^{-1}$ and wind factor of 2% with the new geometry described in Subsection 3.3.6. The inserted plots in panels (a) and (b) compares the salinity and temperature profile (black lines) with their reference profiles at the lead center (x=0 m) and in the control stations located at x=-40 m and x=40 m, respectively.**

**Increasing the separation between leads to a significant reduction in the temperature of the NSTM layer, which is hardly distinguishable from the background thermal field (Figure 15). This is presumed to be due**

**to the scarce feedback between the cores of warm masses generated in distant leads. Warm water patches extend horizontally for a great distance before connecting with the remnants of other warm water patches. This increases mixing with the surrounding environment due in part to a larger sea ice cover on the surface. The result is a significant reduction in the thermal signature.**

I believe that some of the results show here help with these questions, but I would like to see a better justification of the setup.

*The reviewer is referred to the previous answers where this aspect was considered.*

The summary states that "sequence of calm and windy periods in the leads results in a final thermal structure characterized by a spatially distributed NSTM layer" - I'm not sure that this was convincingly demonstrated. Do you *not* get is without the calm periods? A few different scenarios show various levels of patchiness, but the periodic nature of the simulations (when ice is advected) makes it a bit confusing...

*The reviewer is referred to the previous simulations about this subject reported in this document.*

L7: I understand that the focus of the paper is on early summer time, but it might be worth discussing a bit how it connects to winter and seasonality. In particular, while the second sentence of the abstract is probably correct, the heat exchanges are even stronger in the winter. I would probably delete this sentence.

*The sentence is removed as suggested.*

L229: It is a little bit confusing how how is the 'steady wind period' (L226). From Fig 8, it seems that the wind is turned on at the beginning of "day 5" (the sixth day?)

*The day number is assigned to the completed simulation day, that is, the time period from 0-24 hrs refers to day 1, the period from 24-48 hrs refers to day 2 and so on. The following clarification would be added to the caption of Figure 8:*

**Figure 8….The number on the vertical axis corresponds to the days already completed in the simulation.**

L232: Is Fig. 7 a one-day average, or a snapshot?  What does "the circulation on the fifth day is considered" mean? A time average? The next sentence seems to indicate that it is a snapshot in time, but that wasn't clear. Is the circulation (streamlines) calculated from a day-long average?

*It is the final state (snapshot) obtained in the simulation. The following clarifications are included:*

**L 281 A westwards flow was generated by the action of wind and ice movement in the resulting final state of the 10-day simulation (Figure 7a).**

*in the main text and:*

**Similar to Figure 4 but representing the final oceanographic conditions of the water column at the end of the wind period with a wind speed of 6 ms$^{-1}$ and wind factor of 2 %**

*in the figure caption*

L255: 'Reference profiles' and "Initial profiles" are the same and without lateral variability. They should be called the same (it is a bit confusing here what the gray lines were, and if they changed from panels to panels).

*The notation is unified as suggested*

L277: It might be useful to add what day "the end of the simulation period" corresponds to (day 10?), to facilitate interpreting Fig 8 in the context of Fig 7 (and others).

*The sentence is rephrased:*

**L 281 A westwards flow was generated by the action of wind and ice movement in the resulting final state of the 10-day simulation (Figure 7a).**

L 295: The salinity and velocity fields are 'considered' since temperature is effectively a tracer. Rephrase: "Discussion of the results focuses on the impact on the thermal field."

*The sentence is rephrased in the revised manuscript:*

**L 300 The discussion of the results focuses on the impact on the thermal field.**

L300: Specify that the local temperature is warmer, and shallower in this scenario.

*This aspect is specified in the revised manuscript.*

**L302: However, the NSTM layer is warmer and shallower in this scenario.**

L304: The 'control run' has also a depth of 28m, and temperatures seems similar (Fig 7b and Fig 9). This should be captured in the text. Ultimately I think that is the main point here - doubling the FW meltrate doesn't impact the NSTM. Why? It seems that the sensitivity is mostly on using smaller melt rates… More melt doesn't change anything (already isolating the temperature maximum), but less has a large impact… Is the depth-integrated heat content the same?

*The observation is mentioned in the revised manuscript. It is presumed that the states in Figure 7b and 9b are the asymptotic states determine by the depth attenuation of the effects of the surface stress. It would be expected that the case in Figure 9a would approach a similar asymptotic condition for a long enough run. This is because the flow rate of melt waters is smaller and takes longer time to fill the surface layers affected by the surface stress. This hypothesis will be explored and results mentioned in the revised text. The mean heat lost with respect to the final state of the calm simulation is -6.9 MJ/m$^3$, -7.1 MJ/m$^3$ and -7.1 MJ/m$^3$ for cases Figure 9a, Figure 7b and Figure 9b, respectively.*

*The following result has been included in the revised manuscript to clarify this question:*

**L 352 A comparison of Figure 9a with Figures 7b and 9b raises a question about the dissimilarity of the**

**former with the other two. The results in Figures 7b and 9b suggest that over a sufficiently long simulation**

time, the system reaches an intermediate state characterized by an NSTM layer whose structure does not strongly depend on the rate of meltwater inflow but probably on the range of action in the depth of surface stress. Instead, the inflow rate of the meltwater determines the time required to reach the intermediate state. If this were the case, a state similar to that shown in Figures 7b and 9b would result from a simulation of the case with a halved melt rate subject to a wind period greater than 5 days. Figure 10 shows the result obtained when a wind period of 11 days is applied in the case of a halved melt rate. The NSTM layer is now less marked than in Figure 9a but shows a more similar structure with the cases shown in Figures 7b and 9b. In particular, the core of the NSTM layer is now below 20 m deep.

[Figure]

**Figure 10: Similar to Figure 9a but for a model run with 11 days of wind conditions (a total of 16 simulation days including the calm period).**

The last sentence of the abstract states that 'ice drift is key in the development of the NSTM layer'. Closing the lead would do the same, no?

*The reviewer is referred to a previous answer describing the simulation closing the lead during the wind period.*

---

## Editor Decision (ED1)

**REVIEWER I**

The manuscript, "A model for the Artic mixed layer circulation under a summertime lead: Implications on the near-surface temperature maximum formation" by Alvarez develops a simplified 2D model of a summertime Arctic lead using SHEBA data to examine the sensitivity of the near surface temperature maximum to changes in solar radiation, winds, and ice motion.

Global climate models have biased salinity and temperature distributions in the upper Arctic Ocean. It has been suggested this could be associated with resolution or parameterizations of vertical mixing (Rosenblum et al. 2020). Future changes in the upper ocean are also very uncertain as the changing sea ice state may modulate ocean circulation. The manuscript touches on an important issue of simulating processes controlling the structure of the upper Arctic Ocean but requires a clearer discussion of the novel conclusions that separate this study from previous studies of the summertime NSTM and to be put in a broader context in terms of the Arctic as a whole and other more complex models. I have listed these concerns below. My recommendation is major revisions.

*I thank the reviewer for his/her comments and suggestions. I detail below the actions taken to answer or clarify them.*

1. Previous studies, such as Richter-Menge et al. (2001), Steele et al. (2011), and Gallaher et al. (2017) have recognized the importance of solar radiation, winds, and sea ice motion in the formation and persistence of the NSTM layer beyond summer. This is mentioned in the discussion, but the results seem to confirm previous findings, not add very much new. Could you expand on the novel results and the benefits of this model within the hierarchy of the models from the other studies?

   *The following paragraphs (in bold) are suggested for inclusion in the Discussion Section to address this comment:*

L 415 ….A net stretching and deepening of warm waters result from the daily cycle process. ** explains the deepening and lateral spreading under the adjacent ice of warm water masses observed by Richter-Menge et al., (2001) in summertime leads under persistent calm conditions. Furthermore, model results extend the dynamical analysis to subsurface layers not considered in Richter-Menge et al., (2001). The background conditions of these subsurface layers is a fundamental component in the mechanism for the formation of the NSTM layer. Specifically, there is a scientific consensus in attributing the development of the NSTM layer to the solar radiation penetrating in the upper ocean (Maykut and McPhee, 1995; Jackson et al., 2010, Steele et al., 2011, Gallaher et al., 2017). In this study, we find that subsurface heating in summertime leads is also mediated by the sink of warm surface waters to the lower layers by convection cells. The mechanism develops by the buoyancy forcing and lateral boundary conditions of the lead, so it goes unnoticed in one-dimensional study domains (Gallaher et al., 2017) or in open ocean environments (Steele et al., 2011).**

[Figure]

L 433 ….from the sea surface by a layer of cooler, fresher water. **Despite the different model complexity and resolution, Steele et al., (2011, Figure 2a and corresponding text) found the same mechanism deepening the temperature maximum layer in their simulations with the Pan-Arctic Ice–Ocean Modeling and Assimilation System (PIOMAS). On the other hand, the hypothesis attributing the resilience of the NSTM layer to the protective action of the upper layer on the underneath ones (Maykut and McPhee, 1995; Jackson et al., 2010; Gallaher et al., 2017) is not supported here.**

L438 ….and being absent during the wind event

**The two-dimensional physical description of this study highlights other novel aspects related to the formation of the NSTM layer in summer leads. The superficial shear stress generated by the movement of the ice dominates that generated by the wind in the horizontal dispersal of subsurface heat. This is due to the limited wind fetch within the leads. In contrast, in a one-dimensional description of a summer lead physics (Callaher et al., 2017), both sources of surface stress are indistinguishable. The layered structure of the resulting NSTM also emerges from the two-dimensional description used here. Periodic lateral boundary conditions idealize an ice landscape made up of large plates separated by long, narrow open leads as observed during the early melting period (Perovich et al. 2001). The surface stress generated by the repetitive passage of ice plates spreads laterally the warm waters accumulated under the leads, connecting the warm waters accumulated on unconnected leads. Another finding of this study, relates the preferential formation of the NSTM layer in environments with thermal profiles where the temperature decreases with depth. In the opposite case, eroding the top part of the thermal profile after the calm period may only result on a monotonic increase of temperature with depth. This would occur when the temperature just beneath the cold and fresh mixed layer does not exceed the temperature of the lower warm layers. This could be the case in regions of the Eurasian Arctic if warm Atlantic waters flow near the surface.**

*In addition, the revised manuscript will incorporate new and novel parametric analyses of different aspects of the NSTM formation as a result of this reviewing process. These include new initial stratification (see below), the role of the calm period and the effect of the lead separation.*

2.  While there was good discussion about the caveats and limitations of this modeling study, I had a question about a couple more. For example, the model developed here does not include important aspects such as large-scale ocean circulations and only has a very simple representation of freshwater input, which have both been highlighted in the previous studies as important for the NSTM. The model was also developed based on a very short time period from SHEBA and the Arctic has changed considerably since then (Dewey et al., 2018). Does the author expect the conclusions would change with a more recent case? Although it was located in different region of the Arctic, what about with MOSAIC?

*A new sensitivity study using a different and more recent initial profile, would be included to answer this question. In particular, the following section would be incorporated to the study:*

L 361 *3.3.4 Stratification*

**A different initial stratification of the water column was considered in the numerical study to assess the sensitivity of the reported formation mechanism to this variable. Specifically, Figure 12a displays stratification conditions considered for this assessment. The profiles of temperature and salinity were collected in late April of 2015 at 89.20º N, 55º W in the framework of the project North Pole Station: A Distributed Long-Term Environmental Observatory (Kelly and Morison, 2009). Unlike**

[Figure]

**Figure 12: (a) Initial temperature and salinity profiles and (b) thermal distribution at the end of the simulated wind period with wind intensity 6 ms⁻¹ and wind factor of 2%. The inserted plots in panels (a) and (b) compares the salinity and temperature profile (black lines) with their reference profiles at the lead center (x=0 m) and in the control stations located at x=-40 m and x=40 m, respectively.**

**the previous case (Figure 1), almost homogeneous profiles in temperature and salinity characterize the stratification conditions in a large part of the range of depths considered. In addition to homogeneity, the water column is fresher and colder than the previous conditions up to 45 m depth. Below that depth, the temperature profile is warmer than the one observed at SHEBA location (Figure 1). The numerical simulation followed the**

same procedure and forcing reported in Subsections 3.1 and 3.2, except for the  initial conditions of stratification.

A local temperature maximum also develops between 20 and 30 m depth for the selected initial profiles (Figure 12b). The resulting temperature maximum is slightly cooler than that found with the previous stratification (Figure 7b). The NSTM layer clearly differs from the colder upper layer, but the temperature difference between the NSTM and lower layers is less marked. This is a consequence of the fact that the initial temperature profile is warmer at depth than at the surface (contrary to the previously examined thermal profile). The implications of this finding will be discussed below. The spatial variability of the temperature field is smoother than that observed in Figure 7b. This is presumed to be a result of the limited potential energy initially available in the nearly homogeneous density profile (salinity dependent only) down to 20 m depth, to trigger instabilities when surface mixing is active.

*These results would be commented in the Discussion Section:*

Another finding of this study, ==relates the== preferential formation of the NSTM layer in environments  where  temperature decreases with depth. ==In the opposite case,== eroding the top part of the thermal profile after the calm period, may only result on a monotonic increase of temperature with depth. This would occur when the temperature just beneath the cold and fresh mixed layer does not exceed the temperature of the  warm layer. This could be the case in regions of the Eurasian Arctic if warm Atlantic waters flow near the surface.

*The Section will also discuss aspects related to the role of the calm period and the effect of the lead separation.*

3. There are instances throughout the manuscript that require some proofreading. For example, in the paragraph beginning on Line 194, the first and second sentences are missing "are" and "is", respectively and in the last sentence computing should be changed to computing. Another example is in the use of minimum and maximum throughout the manuscript. However, I felt it was well constructed, as I appreciated that the methods section describing the model was easy to follow and the approaches taken are appropriate.

Dewey, S., Morison, J., Kwok, R., Dickinson, S., Morison, D., & Andersen, R. (2018). Arctic ice-ocean coupling and gyre equilibration observed with remote sensing. Geophysical Research Letters, 45(3), 1499-1508.

Rosenblum, E., Fajber, R., Stroeve, J. C., Gille, S. T., Tremblay, L. B., & Carmack, E. C. (2021). Surface salinity under transitioning ice cover in the Canada Basin: Climate model biases linked to vertical distribution of fresh water. Geophysical Research Letters, 48(21), e2021GL094739.

*The revised manuscript will be submitted for professional English editing and proofreading. References will be included in the revised text.*

**REVIEWER II**

This is very interesting work, but I think it's missing a few kew elements to make it significant - I am not convinced that it demonstrate the importance of both the calm and windy periods, as is claimed in the abstract and summary.

I thank the reviewer for his/her comments and suggestions. I detail below the actions taken to answer or clarify them.

My main question is about the run setup, where there is a calm period (5 days) before the wind is turned on. Why is that?

*The ultimate goal of this study is to understand the dynamical mechanism of the generation of the NSTM layer in leads as suggested by previous observations (Maykut and McPhee, 1995; Jackson et al., 2010). The main dynamical components would be then the lead geometry, the buoyancy forcing from ice melting and the penetration of solar radiation in the lead. Richter-Menge et al. (2001) provided observations of the shallowest layers in a lead under these conditions (calm conditions). The authors reported a warm layer of fresh water on the surface of the summer leads is the result of the combined effect of meltwater runoff and solar heating. The authors also noticed that this warm freshwater layer deepens and spreads laterally under adjacent ice if calm conditions persist. So the first questions to address was the dynamical origin of this observation and if this deepening of warm waters could be the mechanism for the formation of the NSTM layer. For this reason, simulations during the calm period were first considered. Numerical results highlighted an interesting dynamics during the calm period. However, this dynamical mechanisms alone could not explain the depth at which the NSTM layer is observed. Other dynamical aspects should be included. Steele et al. (2011) and Callaher et al (2017) related in other ocean environments, the deepening of the warm surface waters with surface stress produced by wind or ice movement. The wind simulations in this work investigate if this is the case for leads too. Specifically, the surface stress should be applied after the surface part of the water column has been heated, for example during the calm period. For guidance and data availability, the work of Richter-Menge et al. (2001) nicely fitted for this scope. This fixed the time scales considered in his study.*

*This rationale is summarized in the revised text in:*

L 81 **This study investigates the dynamics in a summer lead exposed to a sequence of calm and moderate wind conditions such as those reported in Richter-Menge et al. (2001) at ice station SHEBA from July 18th to July 31st 1998. An axisymmetric geometry and a particular thermodynamic forcing are common features to summer leads in calm conditions. For this reason, the study initially focuses on the circulation under a summer lead resulting from the combined effect of lead geometry, solar radiation and sea ice melt. Under these conditions, lateral buoyancy gradients between the edges and center of the lead due to solar heating and ice melt can trigger a circulation in the lead. This circulation would result in a warm layer of fresh water on the surface of the lead. The study is then completed with the analysis of the mixing and deepening, if any, of the warm freshwater layer due to wind events (and associated sea ice drift) after the calm period. The ultimate goal of the study is to assess if these environmental conditions could result in the formation of the NSTM layer as inferred from the observations.**

Once the wind is on, even for the smallest wind speed, a 3 m/s wind with a 2% wind factor, would result in a displacement of 2 km per day for the ice. These add up to reasonably large distances over 5 days (particularly for the larger wind experiments), so the periodic nature of the experiment gets a little strange. The ocean field (for the wind forcing cases) is the results of local (calm) initial conditions and a lot of leads that occasionally input more solar radiation in the ocean. Does it make sense to compare only temperature profiles, if we are not sure if the total heat input has been the same?

*The ice landscape during the early melting season has been described as a large recurring plates separated by long, narrow open leads. This ice landscape has been idealized in this work as large (infinite) sequence of ice-patches and open water leads, that is, a periodic fringe pattern with ice patches and open water leads. All leads share the same geometry and are subjected to the same forcing (wind, solar radiation etc) under this symmetry assumption. This mathematical model is complex enough to highlight the aspects we are interested on but simple enough to easily simulate the combined effect of a large set of recurrent leads. Certainly, it is only an idealization of real ice landscapes where the symmetry tries to capture the recurring distribution of leads. The occurrence of leads with different geometry and at different separations is out of the limits of this mathematical description. Furthermore, numerical simulations would be very computationally demanding with the resolutions employed (tens of centimetres) if cyclic symmetry is not exploited.*

*The following clarifications are considered in the revised manuscript*

L 52 … that ice in the leads melts almost completely by the end of July. **At this time, the ice landscape is made up of large recurring plates separated by long, narrow open leads.**

L 94 … **The ice landscape previously described by recurring plates separated by long, narrow open leads, is idealized here by a large (infinite) fringe pattern with ice-patches and open water leads of 250 m and 50 m wide, respectively, Figure 1.**

Maybe I'm missing something… Is the grid (Fig 3) changing as the "Ice sheets are assumed to move westward with the wind"?

*A Galilean transformation is used to hold the coordinate system at the ice plate while the dynamics of the fluid domain is described in a reference frame in motion with respect to the ice. This numerical technique has been used by other authors in the past to simulate leads in motion (Kozo 1983, Kantha 1995 and Skyllingstad and Denbo 2001). The advantage of the approach is that it does not need to modify the mesh of the domain.*

*It is proposed to add the following paragraph to the revised text to clarify this aspect*

L 97 ….. from July 27th to July 31st. **A Galilean transformation of the system equations and boundary conditions was performed to simulate ice motion under windy conditions (Skyllingstad and Denbo, 2001). With this transformation, the coordinate system is held at the ice plate while the water domain flows past the lead at constant speed as in the lead simulations of Kozo (1983), Kantha (1995) and Skyllingstad and Denbo (2001). In combination with the lateral periodic boundary conditions, this numerical configuration describes the motion of the ice fringe periodic pattern over the water domain. In addition, it keeps the domain geometry unchanged, avoiding remeshing the domain and speeding up numerical simulations.**

The results (and modeling setup) make sense to me for the calm period, but it gets confusing when discussing the wind simulation. In particular, the impact of the initial conditions for the 3 m/s case doesn't make sense to me, since this should be the same in all cases - the lead that created the initial warm surface water (Fig 4b) is 10 km away in the 3 m/s case (Fig. 10a), and 20 km away in Fig 7b. Why would only the 3 m/s case show asymmetry (L300)?

*After the calm period, the mathematical model formally describes an infinite sequence of leads with same underwater oceanography due to the imposed periodic symmetry. When the ice moves, the observer is stationary at the top of the ice  (or lead) represented by the computational domain. As a consequence of the periodic symmetry and the relative motion of the ice, the observer on the ice  observes the water masses of the leads in the fringe pattern advected past the observer location. Water masses leave the domain at the eastern boundary while the water masses of the next lead enter in the domain through the western boundary. The oceanographic fields evolve during each cycle due to the surface stress generated by the ice motion. In the first cycle, the observed would observe the water masses of the nearest lead perturbed by the pass of one ice stripe, in the second cycle the observed would find the water masses of the second closest lead perturbed by the pass of two ice stripes and so on.*

*The initial cores of water masses spread out horizontally with the number of cycles. This is just a consequence of the surface stress generated by the ice movement. After some time, the initial cores of warm waters are not anymore individually distinguishable but form a more or less homogeneous thermal layer. The number of cycles to observe the layer structure depends on the intensity of the shear stress. The asymmetry in Fig 10a reveals that more than 5 days with wind conditions are required to homogenise the original thermal field for the case of 3 ms$^{-1}$.*

It might make more sense to 'close the leads' at the beginning of the wind period, to decouple these two processes (adding more heat or freshwater at the surface), and adding mixing. Ultimately I'm not entirely sure what the simulations demonstrate as they are setup now.

*The numerical simulation suggested by the reviewer was done closing the leads at the beginning of the wind period. The Figure below shows the formation at the end of the 10-day simulation (the usual 5-day with lead and calm conditions plus the 5-day of wind conditions and closed lead). The NSTM layer is cooler than that of Figure 7b due to the increase of shear stress (larger ice surface) and melt water input. Still the simulation shows the resilience of the NSTM feature due to its relative isolation from the surface.*

[Figure]

**Figure: Thermal distribution at the end of a 5-day simulated wind period with wind intensity 6 ms$^{-1}$ and wind factor of 2% with closed leads.**

Some of the big questions that this study could really help answer are:

- Do we need calm conditions (ice not moving and lead being open for a long time) to get enough heat in the surface layer to get an NSTM?

[Figure]

**Figure 12a**

*The following results will be included in the revised manuscript to attempt to answer this comment*

**3.3.5 Absence of the calm period**

**New numerical simulations evaluate the relevance of the calm period as a conditioning phase for the formation of the NSTM layer. Wind and ice movement are activated from the start. Thus, the numerical setup skips the 5 days of calm conditions. In these runs, the model was run during 5 days forced as detailed in Subsection 3.2. The profiles displayed in Figures 1 and 12a were considered for initialization.**

**Figure 13a shows the formation of a layer with a temperature maximum when the stratification displayed in Figure 1 initializes the model. The depth and structure of the emerging NSTM layer resembles that of Figure 7b where the calm period was**

[Figure]

**Figure 13: Thermal distribution at the end of a 5-day simulated wind period with wind intensity 6 ms⁻¹ and wind factor of 2% without considering the calm period for the initialization displayed in (a) Figure 1 and (b) Figure 12a. The inserted plots in panels (a) and (b) compares the salinity and temperature profile (black lines) with their reference profiles at the lead center (x=0 m) and in the control stations at x=-40 m and x=40 m, respectively.**

considered. In this case, however, the value of the temperature of the NSTM layer is lower than that formed after the calm period. At the NSTM layer depth, the root mean square difference of temperature is about 0.02ºC with local differences up to almost 0.4 ºC between both cases. The result can be interpreted in terms of erosion of the relatively warm surface part of the initial profile (Figure 1) by the surface stress. In this circumstance, the heat accumulated during the calm period adds to the initial conditions to increase the value of the temperature maximum.

The profile described in Figure 1 resulted at the SHEBA station after a relatively long period of gentle wind conditions (Ritchter-Menge et al., 2001). Thus, this profile could accumulate the warming resulting from a longer calm period. If true, the relevance of the conditioning phase could be underestimated in the previous experiment. A second model run considered an initial stratification that was not suspected to be subject to previous surface heating, such as the profile analyzed in 3.3.4 and shown in Figure 12a. A monotonic increase of temperature with depth is obtained after the 5-day simulation for this profile, Figure 13b. A very shallow and local temperature maximum appears at the trailing edge of the lead. Its dynamical origin is not related with the scope of this study. As in other simulations, it is generated by the advection of surface waters by the wake of the lead. Comparison between Figures 12b and 13b shows that the existence of a calm period is required for the formation of the NSTM layer in this case.

- Do we need (large?) wind events to mix this water to depths comparable to what we observe?

*Calm conditions produce a very localized at the lead's center and near surface (less than 10 m depth) temperature maximum. Numerical results (Figure 8) suggest that mixing by wind or ice movement is required to deepen the temperature maximum to the observed depths (10 to 30 m depth depending of the location). This agrees with the results from Steele et al. (2011) derived for open sea conditions. However, large wind events or moderate winds for long time period could prevent the formation of the NSTM. In the current simulation the thermal signature of the NSTM is almost unnoticed for wind speeds of 9 ms$^{-1}$. That is, the generation and deepening of the NSTM layer seem occurs under certain range of mixing intensity. The following paragraph would be added to the revised manuscript:*

*L 424 …* **The NSTM layer survives under moderate wind and associated ice motion, disappearing for wind intensities greater than 9 ms$^{-1}$. This suggests that the formation of the NSTM layer in leads occurs under certain range of mixing intensity. A high mixing rate resulting from surface stress would lead either to a homogenization of the water column or to a monotonic dependence of temperature with depth, avoiding the appearance of a temperature maximum.**

According to the numerical simulations

- how quickly does the 'patchiness' of the formation (leads) disappear? If one was to increase the separation between leads, would that make the NSTM more patchy?

*The disappearance of the patchiness depends on the intensity of the applied surface stress. For the main case (Figure 7b) the patchiness structure disappears after 3 days of applying the wind forcing and the ice motion. Instead, the patchiness of warm waters*

*would take much longer to disappear if the ice does not move (Figure 11a). Regarding the second question, the following simulation will be added to the revised manuscript:*

**3.3.6 Lead separation**

**The geometry used in this study is representative of the observed main lead fraction. The separation between leads establishes the distance between the cores of warm water masses accumulated under the recurring leads during the calm period. The lateral spreading induced by the surface stress connects the remains of the initial warm water cores. The next sensitivity study considers the evolution of the leads system when the distance between leads is almost double than in the main geometry considered up to now. Specifically, the width of the ice plates separating the leads is 450 m instead the previous width of 250 m. The width of the leads remains the same as in previous simulations. The new geometry was meshed using the same procedure as detailed in sub-Section 2.2, resulting this time in 11736 nodes that generate 22839 triangles. The highest spatial resolution is under the lead similarly to the previous cases. Simulations for the calm and wind period for the new geometric configuration followed the procedure and forcing reported in Subsections 3.1 and 3.2.**

[Figure]

**Figure 13: Thermal distribution at the end of the simulated wind period with wind intensity 6 ms$^{-1}$ and wind factor of 2% with the new geometry described in Subsection 3.3.6. The inserted plots in panels (a) and (b) compares the salinity and temperature profile (black lines) with their reference profiles at the lead center (x=0 m) and in the control stations located at x=-40 m and x=40 m, respectively.**

 **increasing the separation between leads to a significant reduction of the temperature of the NSTM layer, hardly distinguishable from the background thermal field (Figure 13). This is presumed to be due to the scarce feedback between the cores of warm masses generated in distant leads. Warm water patches extend horizontally for a great distance before connecting with the remnants of other warm water patches. This increases mixing with the surrounding environment due in part to a larger ice sheet on the surface. The result is a significant reduction of the thermal signature.**

I believe that some of the results show here help with these questions, but I would like to see a better justification of the setup.

*The reviewer is referred to the previous answers where this aspect was considered.*

The summary states that "sequence of calm and windy periods in the leads results in a final thermal structure characterized by a spatially distributed NSTM layer" - I'm not sure that this was convincingly demonstrated. Do you *not* get is without the calm periods? A few different scenarios show various levels of patchiness, but the periodic nature of the simulations (when ice is advected) makes it a bit confusing...

*The reviewer is referred to the previous simulations about this subject reported in this document.*

L7: I understand that the focus of the paper is on early summer time, but it might be worth discussing a bit how it connects to winter and seasonality. In particular, while the second sentence of the abstract is probably correct, the heat exchanges are even stronger in the winter. I would probably delete this sentence.

*The sentence will be removed as suggested.*

L229: It is a little bit confusing how how is the 'steady wind period' (L226). From Fig 8, it seems that the wind is turned on at the beginning of "day 5" (the sixth day?)

*The day number is assigned to the completed simulation day, that is, the time period from 0-24 hrs refers to day 1, the period from 24-48 hrs refers to day 2 and so on. The following clarification would be added to the caption of Figure 8:*

**Figure 8…. The number on the vertical axis corresponds to the days already completed in the simulation.**
[Figure]

L232: Is Fig. 7 a one-day average, or a snapshot? What does "the circulation on the fifth day is considered" mean? A time average? The next sentence seems to indicate that it is a snapshot in time, but that wasn't clear. Is the circulation (streamlines) calculated from a day-long average?

*It is the final state (snapshot) obtained in the simulation. The following clarifications are proposed:*

**A westward flow has been generated by the action of wind and ice movement in the resulting final state of the 10-day simulation (Figure 7a)**

*in the main text and:*

**Similar to Figure 4 but representing the final oceanographic conditions of the water column at the end of the wind period with a wind speed of 6 ms⁻¹ and wind factor of 2 %**

*in the figure caption*

L255: 'Reference profiles' and "Initial profiles" are the same and without lateral variability. They should be called the same (it is a bit confusing here what the gray lines were, and if they changed from panels to panels).

*The notation will be unified as suggested*

L277: It might be useful to add what day "the end of the simulation period" corresponds to (day 10?), to facilitate interpreting Fig 8 in the context of Fig 7 (and others).

*It is suggested to rephrase the sentence:*

**A westward flow has been generated by the action of wind and ice movement in the resulting final state of the 10-day simulation (Figure 7a)**

L 295: The salinity and velocity fields are 'considered' since temperature is effectively a tracer. Rephrase: "Discussion of the results focuses on the impact on the thermal field."

*The sentence will be rephrased in the revised manuscript.*

L300: Specify that the local temperature is warmer, and shallower in this scenario.

*This aspect will be specified in the revised manuscript.*

L304: The 'control run' has also a depth of 28m, and temperatures seems similar (Fig 7b and Fig 9). This should be captured in the text. Ultimately I think that is the main point here - doubling the FW meltrate doesn't impact the NSTM. Why? It seems that the sensitivity is mostly on using smaller melt rates… More melt doesn't change anything (already isolating the temperature maximum), but less has a large impact… Is the depth-integrated heat content the same?

*The observation will be mentioned in the revised manuscript. It is presumed that the states in Figure 7b and 9b are the asymptotic states determine by the depth attenuation of the effects of the surface stress. It would be expected that the case in Figure 9a would approach a similar asymptotic condition for a long enough run. This is because the flow rate of melt waters is smaller and takes longer time to fill the surface layers affected by the surface stress. This hypothesis will be explored and results mentioned in the revised text. The mean heat lost with respect to the final state of the calm simulation is -6.9 $MJ/m^3$, -7.1 $MJ/m^3$ and -7.1 $MJ/m^3$ for cases Figure 9a, Figure 7b and Figure 9b, respectively.*

The last sentence of the abstract states that 'ice drift is key in the development of the NSTM layer'. Closing the lead would do the same, no?

*The reviewer is referred to a previous answer describing the simulation closing the lead during the wind period.*

---

## Author Response (AR2)

**ANSWER TO REFEREE II**

I like the changes that the author has made. The paper is clearer. I think it is an interesting set of results, and it would be good to punish this work.

*I thank the comments and suggestion of the reviewer to improve the manuscript. The text has been modified according to the comments in the way detailed below.*

The experiment shown in Fig 10 (running the wind period for longer) suggest that these are not steady state results. This somewhat contradicts the plots of the evolution of the temperature and depth of the local maximum (Fig 8 - which I initially interpreted as converging, but perhaps they are simply steadily and slowly cooling/deepening?). I am struggling a bit to understand the implications for the real world. I guess one could argue that we always see calm periods followed by windy periods, but I'm not entirely sure how to think of all these variables and represent the somewhat stable and widely distributed NSTM that we observe. This is a bit different than the discussion of the 'layered' structure (L612)

This concern shouldn't prevent the publication, but perhaps the summary should discuss this a bit.

*It is now mentioned in line 617: The location of the NSTM layer in the water column converges to depth values where eroding surface mechanisms (mixing and surface stress) are not effective, with only diffusion remaining as a thermal homogenization process. The latter requires longer time scales than those considered in this study.*

*The study investigates a mechanism for the formation of the NSTM but does not exclude other possibilities. The observed NSTM could result from the contribution of diverse formation processes. This aspect is mentioned in L 629:*

*While the present results show that the sequence of calm and windy/ice motion periods in summer leads to the formation of an NSTM layer, this local mechanism does not exclude other processes or locations. Steele et al. (2011) reported the generation of the NSTM layer at the basin scale, where large-scale ocean dynamics play a significant role. Therefore, certain combinations of dynamic mechanisms, either on a local or global scale, can result in the formation of NSTMs.*
* * *
L9: It's a bit unclear what "this period" refers to, since previous sentences are about both winter and summer. Just write "summer"

*L9 It is now written: remain largely unexplored during summertime*

L50: Except in this paragraph, the definition of "lead" is an ice-free area. It might be better to stick with this, and specify throughout this paragraph that the author refers to "refrozen leads" here.

*L50 it is now written: During the early melting season, refrozen leads become preferential melting sites*

L353: I think it would be useful here for the text to explicitly refer to what runs Fig 9a (half melt rate) and Fig 9b (double) represent, as opposed to just the figure. It is confusing (particularly the sentence on L353).

*L353 it is now written: A comparison of Figure 9a (run with half melt rate) with Figures 7b and 9b (run with double melt rate) raises*

**The same comment applies to the caption of Fig 14, for example. Please state what these simulations are, it will make the paper easier to read.**

*It is now written in Caption Figure 14: Thermal distribution at the end of a 5-day simulated wind period with wind intensity 6 ms-1 and wind factor of 2% without considering the calm period. The upper panel (a) shows the results when the run is initialized with the profile shown in Figure 1. The lower panel (b) display the temperature field when the run is initialized with the profile shown in Figure 12a.*